# Mapping kinase domain resistance mechanisms for the MET receptor tyrosine kinase via deep mutational scanning

**Gabriella O Estevam[1,2], Edmond Linossi[3,4], Jingyou Rao[5], Christian B Macdonald[1], Ashraya Ravikumar[1], Karson M Chrispens[1,6], John A Capra[7], Willow Coyote-Maestas[1,8], Harold Pimentel[5,9,10], Eric A Collisson[11,12], Natalia Jura[3,4,8], James S Fraser[1,8]***

[1]Department of Bioengineering and Therapeutic Sciences, University of California, San Francisco, San Francisco, United States; [2]Tetrad Graduate Program, University of California, San Francisco, San Francisco, United States; [3]Cardiovascular Research Institute, University of California, San Francisco, San Francisco, United States; [4]Department of Cellular and Molecular Pharmacology, University of California, San Francisco, San Francisco, United States; [5]Department of Computer Science, University of California, Los Angeles, Los Angeles, United States; [6]Biophysics Graduate Program, San Francisco, United States; [7]Bakar Computational Health Sciences Institute and Department of Epidemiology and Biostatistics, University of California, San Francisco, San Francisco, United States; [8]Quantitative Biosciences Institute, University of California, San Francisco, San Francisco, United States; [9]Department of Computational Medicine and Human Genetics, University of California, Los Angeles, Los Angeles, United States; [10]Department of Human Genetics, David Geffen School of Medicine, University of California, Los Angeles, Los Angeles, United States; [11]Human Biology, Fred Hutchinson Cancer Center, Seattle, United States; [12]Department of Medicine, University of Washington, Seattle, United States

**\*For correspondence:**
jfraser@fraserlab.com

## eLife Assessment

This manuscript provides an **important** overview of potential resistance mutations within MET Receptor Tyrosine Kinase. The evidence supporting the findings is **convincing** - it should be pointed out that the approach is comparatively new for the application of protein kinases and the results are therefore of potentially great value. The results will be of value for clinicians facing drug resistance mutations, computational biologists who are training models of drug resistance mechanisms and biologists with an interest in cell signaling.

**Abstract** Mutations in the kinase and juxtamembrane domains of the MET Receptor Tyrosine Kinase are responsible for oncogenesis in various cancers and can drive resistance to MET-directed treatments. Determining the most effective inhibitor for each mutational profile is a major challenge for MET-driven cancer treatment in precision medicine. Here, we used a deep mutational scan (DMS) of ~5764 MET kinase domain variants to profile the growth of each mutation against a panel of 11 inhibitors that are reported to target the MET kinase domain. We validate previously

identified resistance mutations, pinpoint common resistance sites across type I, type II, and type I ½ inhibitors, unveil unique resistance and sensitizing mutations for each inhibitor, and verify non-cross-resistant sensitivities for type I and type II inhibitor pairs. We augment a protein language model with biophysical and chemical features to improve the predictive performance for inhibitor-treated datasets. Together, our study demonstrates a pooled experimental pipeline for identifying resistance mutations, provides a reference dictionary for mutations that are sensitized to specific therapies, and offers insights for future drug development.

## Introduction

Receptor Tyrosine Kinases (RTKs) are critical signaling molecules that activate and regulate cellular pathways. Disruption of typical RTK regulatory mechanisms through point mutations, gene amplification, protein fusions, or autocrine loops can drive the development, maintenance, and spread of cancers. Small molecule inhibitors are designed to disrupt aberrant signaling cascades by selectively targeting the kinase domain, with tyrosine kinase inhibitors (TKIs) like imatinib showing durable treatment outcomes (*Cohen et al., 2021*; *Attwood et al., 2021*). Inhibitors are generally designed against either a wild-type kinase or a specific mutational profile, yet acquired mutations can alter sensitivity to different inhibitors and undermine efficacy. The most extreme case of this is resistance, which emerges in the treatment of many cancers by TKI selective pressure (*Cohen et al., 2021*; *Attwood et al., 2021*). These mutations may act by altering kinase stability, expression, conformation, or activity of the target kinase. Although several recurrent mutations at inhibitor-interacting positions have predictable resistance, specific and rare resistance mutations can be associated with the interactions or conformations unique to certain inhibitors.

An attractive strategy to counter resistance is optimizing the interactions that differ between inhibitors. Small-molecule kinase inhibitors fall into four distinct groups, characterized by their binding modality to the ATP pocket and conformational preferences (*Arter et al., 2022*; *Attwood et al., 2021*; *Zuccotto et al., 2010*). Among these groups, three are ATP-competitive: type I, type II, and type I½ (*Figure 1A–C*). Type I inhibitors occupy the adenosine binding pocket, form hydrogen bonds with 'hinge' region residues, and favor an active conformation. Type II inhibitors also occupy the adenosine pocket but extend into an opening in the R-spine that is accessible in an inactive conformation (*Arter et al., 2022*; *Figure 1B*). Type I½ inhibitors combine features from both type I and type II inhibitors, engaging with both the adenosine pocket and the R-spine pocket (*Arter et al., 2022*; *Figure 1B*). Finally, type III inhibitors are allosteric, non-ATP competitive inhibitors (*Arter et al., 2022*; *Figure 1B*). Given the chemical interaction differences and conformational preferences among TKI groups for distinct kinase states, a general approach to combating resistance is sequential treatment of type I and II inhibitors (*Recondo et al., 2020a*). However, without understanding the potential sensitivity of an acquired resistance mutation to the subsequent inhibitor, the efficacy of such strategies is not guaranteed.

The problem of which inhibitor to use and in what order is exemplified by the choice of inhibitors targeting MET kinase (*Recondo et al., 2020b*; *Fernandes et al., 2021*). MET is an RTK and proto-oncogene that has been implicated in the pathogenesis of gastric, renal, colorectal, and lung cancers (*Frampton et al., 2015*; *Duplaquet et al., 2018*; *Wood et al., 2021*; *Lu et al., 2017*). Molecular profiling and next-generation sequencing of tumor samples has provided insight on cancer-associated MET variants (*Frampton et al., 2015*; *Bahcall et al., 2022*). Clinical reports following post-treatment outcomes have documented recurrent resistance mutations at positions such as D1228, Y1230, G1163, L1195 for MET (*Fernandes et al., 2021*; *Lu et al., 2017*; *Recondo et al., 2020a*; *Li et al., 2017*). The challenge of acquired resistance following MET inhibitor therapy has been approached with strategies including sequential treatment of type I and type II TKIs (*Recondo et al., 2020b*; *Bahcall et al., 2016*, *Cai et al., 2021*), and combination therapy with type I and type II TKIs (*Bahcall et al., 2022*; *Fernandes et al., 2021*; *Smyth et al., 2014*). However, without extensive documentation of the behavior of resistance and sensitizing mutations in MET towards specific TKIs, there remains a barrier towards leveraging inhibitors for specific mutational responses, optimizing inhibitor pairings, and informing rational drug design. Thousands of compounds have been screened against the MET kinase domain, and while several have undergone clinical trials, currently four MET inhibitors have received FDA approval: crizotinib, cabozantinib, tepotinib, and capmatinib (*Santarpia et al., 2021*;

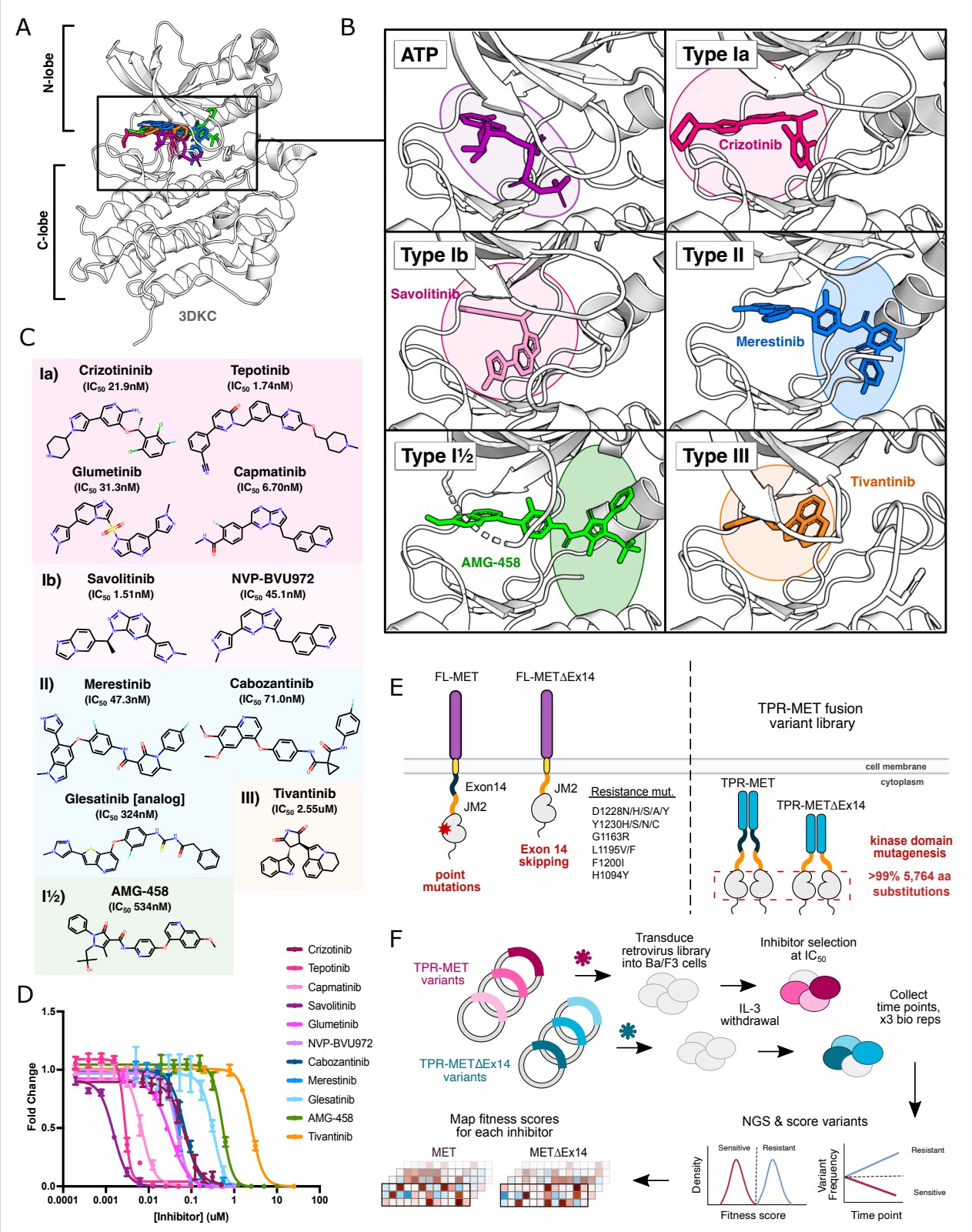

**Figure 1.** MET kinase inhibitor types and resistance mutations screened against a nearly comprehensive library of kinase domain substitutions. (**A**) Crystal structure of the ATP-bound MET kinase domain (3DKC) overlaid with type Ia (crizotinib, 2WGJ), type Ib (savolitinib, 6SDE), type II (merestinib, 4EEV), type I½ (AMG-458, 5T3Q), and type III inhibitors (tivantinib, 3RHK). (**B**) Pocket view of ATP and each inhibitor type bound to the active site of the MET kinase domain with the respective inhibitor and crystal structures from panel A. (**C**) 2D chemical structures of each inhibitor screened against

*Figure 1 continued on next page*

*Figure 1 continued*

the site saturation mutagenesis library of the MET kinase domain, with each experimentally determined IC$_{50}$ values displayed for Ba/F3 cells stably expressing the wild-type MET ICD in a TPR-fusion background. (**D**) Dose-response curves for each inhibitor against the wild-type MET intracellular domain expressed in a TPR-fusion in the Ba/F3 cell line (n=3). (**E**) Schematics of the full-length and exon 14 skipped MET receptor alongside the TPR-fusion constructs with the full-length and exon 14 skipped intracellular domain, displaying four mechanisms of oncogenic activity: point mutations, exon 14 skipping, constitutive activity through domain fusions, and inhibitor resistance mutations. (**F**) Experimental workflow for defining the mutational landscape of the wild-type TPR-MET and exon 14 skipped TPR-METΔEx14 intracellular domain against 11 ATP-competitive inhibitors in Ba/F3, interleukin-3 (IL-3) withdrawn pooled competition assay.

The online version of this article includes the following figure supplement(s) for figure 1:

**Figure supplement 1.** Structural inhibitor classification and dose-response determination.

**Figure supplement 2.** Correlation analysis of the MET kinase domain site saturation mutagenesis library across replicates and conditions.

**Figure supplement 3.** Correlation analysis of the METΔEx14 kinase domain site saturation mutagenesis library across replicates and conditions.

**Figure supplement 4.** Fitness landscapes of the MET kinase domain against a panel of 11 inhibitors.

*Wang and Lu, 2023*). Nevertheless, the emergence of resistance not only limits the efficacy of these drugs but also poses challenges for second-line therapeutic strategies, particularly in the context of rare and novel mutations.

Previously, we used deep mutational scanning (DMS), a pooled cellular selection experiment, to massively screen a library of nearly all possible MET kinase domain mutations. The juxtamembrane domain is encoded by exon 14 in MET, serves an incompletely understood negative regulatory function in the kinase (*Ma et al., 2003*) and is recurrently excluded in cancer by somatically encoded exon skipping mutations. By testing this library in the context of a wild-type intracellular domain and the recurrent cancer exon 14 skipped variant (METΔEx14; *Figure 1E*), we identified conserved regulatory motifs, interactions involving the juxtamembrane and αC-helix, a critical β5 motif, clinically documented cancer mutations, and classified variants of unknown significance (*Estevam et al., 2024*). Understanding how these variants respond to specific inhibitors can inform therapeutic strategies, with precedent in inhibitor-based DMS studies across kinases such as ERK, CDK4/6, Src, EGFR, and others (*Brenan et al., 2016*; *Persky et al., 2020*; *Chakraborty et al., 2024*; *An et al., 2023*).

Here, we explore the landscape of TKI resistance of the MET kinase domain against a panel of 11 inhibitors, utilizing our previously established platform (*Estevam et al., 2024*). By profiling a near-comprehensive library of kinase domain variants in the MET and METΔEx14 intracellular domain, we captured a diverse range of effects based on inhibitor chemistry and 'type' classifications (*Figure 1*). Within our screen, mutations that confer resistance and offer differential sensitivities across inhibitors were identified, which can be leveraged in sequential or combination therapy. We use Rosace, a Bayesian fitness scoring framework, to reduce false discovery rates in mutational scoring and allow for post-processing normalization of inhibitor treatments (*Rao et al., 2024*). With our dataset, we have analyzed differential sensitivities to inhibitor pairs and provided a platform for assessing inhibitor efficacy based on mutational sensitivity and likelihood. Lastly, we augment a protein language model (*Rives et al., 2021*; *Brandes et al., 2023*; *Chen and Guestrin, 2016*) with biophysical and chemical features to improve predictions for MET inhibitor datasets, and in the future more effectively learn and predict mutational fitness towards novel inhibitors.

## Results

### Measuring the mutational fitness of 5,764 MET kinase domain variants against ATP-competitive inhibitors

To evaluate the response of MET mutations to different inhibitors, we selected six type I inhibitors (crizotinib, capmatinib, tepotinib, glumetinib, savolitinib, and NVP-BVU972), three type II inhibitors (cabozantinib, glesatinib analog, merestinib), and a proposed type III inhibitor, tivantinib (*Figure 1C*). Type I MET inhibitors leverage pi-stacking interactions with Y1230 and salt-bridge formation between D1228 and K1110, and are further classified as type Ia or Ib based on whether they interact with solvent front residue G1163 (*Cui, 2014*; *Fujino et al., 2019*; *Wang and Lu, 2023*; *Figure 1C*). Here, we specifically define type Ia inhibitors as having a solvent-front interaction (*Cui, 2014*), which structurally classifies tepotinib and capmatinib as type Ia based on our analysis of experimental structures

and inhibitor docked models (*Figure 1A–C*; *Figure 1—figure supplement 1*), despite classification as type Ib in other studies (*Brazel et al., 2022*; *Fujino et al., 2022*; *Recondo et al., 2020a*).

As in our previous work, we employed the Ba/F3 cell line as our selection system due to its undetectable expression of endogenous RTKs and addiction to exogenous interleukin-3 (IL-3; *Estevam et al., 2024*). These properties allow for positive selection based on ectopically expressed kinase activity and proliferation in the absence of IL-3 (*Daley and Baltimore, 1988*; *Warmuth et al., 2007*; *Koga et al., 2022*). We used a TPR-MET fusion to generate IL-3-independent constitutive activity (*Estevam et al., 2024*). While this system affords cytoplasmic expression, constitutive oligomerization, and HGF-independent activation, features like membrane-proximal effects are lost (*Cooper et al., 1984*; *Park et al., 1986*; *Peschard et al., 2001*; *Rodrigues and Park, 1993*; *Vigna et al., 1999*; *Mak et al., 2007*; *Pal et al., 2017*; *Lu et al., 2017*; *Fujino et al., 2019*). The constitutive activity of TPR-MET and the reliance of Ba/F3 cells on that activity render this selection system quite sensitive for determining reduction in growth from small molecule inhibition.

We generated dose response curves for each inhibitor against wild-type TPR-MET (wild-type intracellular domain, including exon 14) and TPR-METΔEx14 (exon 14 skipped intracellular domain) constructs, stably expressed in Ba/F3 cells to determine $IC_{50}$ values for our system (*Figure 1D*; *Figure 1—figure supplement 1*). We used our previously published library harboring >99% of all possible 5764 kinase domain (1059-1345aa) mutations in a TPR-fusion background carrying either a wild-type MET or exon 14 skipped intracellular domain (*Estevam et al., 2024*; *Figure 1E*). Time points were selected every two cell doublings over the course of four time points, and cells were split and maintained in the absence of IL-3 and presence of drug at $IC_{50}$ for each inhibitor, including a DMSO control. All samples, across all time points and replicates, were prepared for next-generation sequencing (NGS) in parallel, and sequenced on the same Illumina NovaSeq 6000 flow cell to identify variant frequencies (*Figure 1F*). We then calculated variant fitness scores using Rosace (*Rao et al., 2024*; *Figure 1F*; *Figure 1—figure supplements 1–4*). We performed parallel analysis of the TPR-MET and TPR-METΔEx14 screens; however, we focus our analysis below on TPR-MET with the parallel and largely consistent analyses of TPR-METΔEx14 available in the supplement.

## Defining the mutational landscape of resistant and sensitizing mutations for the MET kinase domain

While growth rates were experimentally controlled through equipotent dosing during selection, there was no direct way to validate this post-processing. To generate meaningful comparisons between inhibitor scores and conditions, in addition to performing downstream score subtractions, we normalized cell growth rates for each inhibitor to the growth rate observed for the DMSO population. As expected, the DMSO control population displayed a bimodal distribution with mutations exhibiting wild-type fitness centered around 0, with a wider distribution of mutations that exhibited loss- or gain-of-function effects, as defined by fitness scores with statistically significant lower or greater scores than wild-type, respectively (*Figure 2A*; *Figure 2—figure supplement 1*). Also as expected, inhibitor-treated populations displayed distributions with a loss-of-function peak, representative of mutations that are sensitive to the inhibitor. Unlike DMSO, inhibitor populations were right-skewed, showing greater enrichment of gain-of-function scores at the positive tail of distributions (*Figure 2A*). These population differences were exemplified by the low correlation between each inhibitor and DMSO, with capmatinib showing the greatest difference from DMSO with a Pearson's correlation of 0.45, and tivantinib standing as an outlier with a Pearson's correlation of 0.93 (*Figure 2B*).

In comparing all conditions to each other, we were able to further capture differences between and within inhibitor types. Type II inhibitors displayed the greatest similarities to one another, with merestinib and the glesatinib analog having the highest correlation (*r*=0.93) and cabozantinib and glesatinib analog showing the lowest (*r*=0.87; *Figure 2C*). While type I inhibitors were also highly correlated, capmatinib stood out as an outlier, displaying the lowest correlations potentially due to difficulty in experimental overdosing due to its greater potency (*Bahcall et al., 2022*; *Fujino et al., 2019*; *Figure 2C*). While there was only one type I½ inhibitor, AMG-458, it displayed higher similarity to type II inhibitors than to type I, likely due to similar type II back pocket interactions with the kinase R-spine (*Figure 1B*; *Figure 2C*). Nevertheless, AMG-458 was most distinct from cabozantinib (*r*=0.83) and type I inhibitors tepotinib (*r*=0.79) and savolitinib (*r*=0.73; *Figure 2C*). Between type I and type

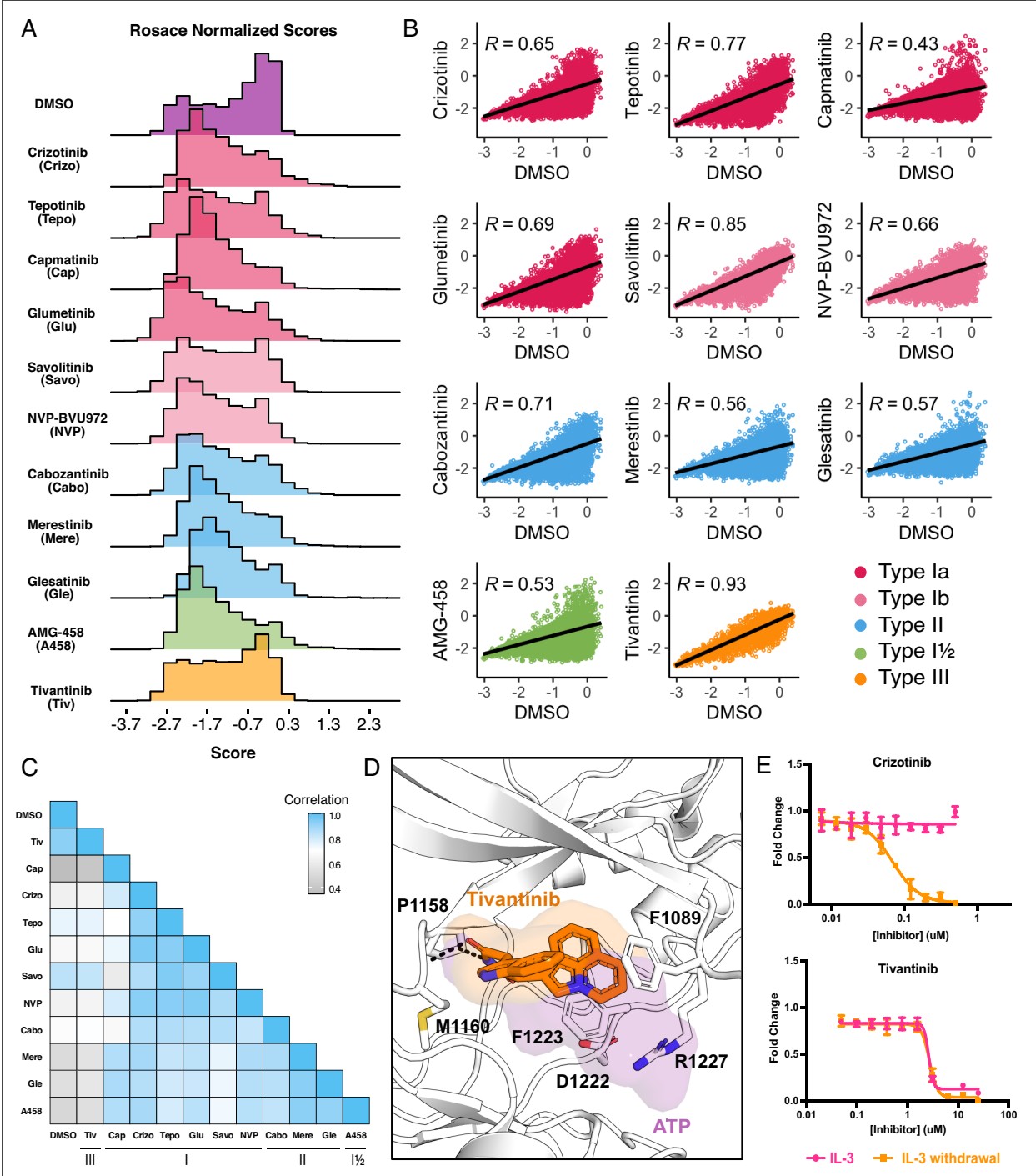

**Figure 2.** Mutational landscape of the MET kinase domain under 11 ATP-competitive inhibitor selection. (**A**) Distributions of all variants (wild-type synonymous, early stop, and missense) for each condition in the wild-type TPR-MET kinase domain, scored with Rosace and normalized to the growth rate of the DMSO control population. (**B**) Correlation plots for all mutational fitness scores for each drug against DMSO, fitted with a linear regression and Pearson's R value displayed. (**C**) Heatmap showing the Pearson's R correlation for each condition against each other, annotated by condition and inhibitor type. Correlations are colored according to a scale bar from gray to blue (low to high correlation). (**D**) Crystal structure of the tivantinib-bound MET kinase domain (PDB 3RHK) overlaid with the ATP-bound kinase domain (PDB 3DKC), with tivantinib-stabilizing residues and overlapping density of tivantinib (orange) and ATP (purple) highlighted. (**E**) Dose responses of crizotinib and tivantinib tested against stable Ba/F3 cells expressing the wild-type intracellular domain of MET fused to TPR, tested in the presence and absence of interleukin-3 (IL-3) (n=3).

The online version of this article includes the following figure supplement(s) for figure 2:

**Figure supplement 1.** Mutational landscape of the METΔEx14 kinase domain under 11 ATP-competitive inhibitor selection.

II groups, with the exception of capmatinib, tepotinib, and savolitinib showed the lowest correlation with merestinib and glesatinib analog (*Figure 2C*).

The strong correlation of tivantinib with the DMSO control (*r*=0.93) and low correlation with all other inhibitors suggested a MET-independent mode of action. Until recently (*Michaelides et al., 2023*), tivantinib was considered the only type III MET-inhibitor and showed promising early clinical trial results (*Eathiraj et al., 2011*; *Bahcall et al., 2022*). In vitro assays on the purified MET kinase domain have shown that tivantinib has the potential to hinder catalytic activity (*Munshi et al., 2010*) and structural studies revealed that it selectively targets an inactive DFG-motif conformation, with tivantinib stabilizing residues (F1089, R1227) blocking ATP binding (*Eathiraj et al., 2011*; *Figure 2D*). Yet in contradiction, comparative MET-dependent and MET-independent cell-based studies on tivantinib have also shown MET agnostic anti-tumor activity, posing that tivantinib may have an alternative inhibitory mechanism than an MET-selective one (*Michieli and Di Nicolantonio, 2013*; *Basilico et al., 2013*; *Katayama et al., 2013*; *Fujino et al., 2019*).

Therefore, to test the hypothesis that tivantinib is not MET-selective in our system, we compared the dose response of tivantinib and crizotinib in the presence and absence of IL-3 for wild-type TPR-MET, stably expressed in Ba/F3 cells (*Figure 2E*). As expected, crizotinib only displayed an inhibitory effect under IL-3 withdrawal, highlighting a MET-dependent mode of action. In contrast, tivantinib displayed equivalent inhibition regardless of IL-3, reinforcing that tivantinib has cytotoxicity effects unrelated to MET inhibition (*Figure 2E*) and underscores the sensitivity of the DMS in identifying direct protein-drug effects.

## Crizotinib-MET kinase domain resistance profiles exemplify the information accessible from individual inhibitor DMS

As an example of the insights that can be learned from the inhibitor DMS screens, we examined the profile for crizotinib, one of four FDA approved inhibitors for MET and a multitarget TKI (*Cui et al., 2011*; *Wang and Lu, 2023*; *Santarpia et al., 2021*). To identify mutations that show gain-of-function and loss-of-function behaviors specific to inhibitors compared to DMSO, we subtracted DMSO from all fitness scores. (*Figure 3A*), with the expectation that effects related to expression or stability would be similar in both conditions, enhancing the ability to identify drug sensitivity or resistance. Indeed, the highest frequency of gain-of-function mutations occurred at residues mediating direct drug-protein interactions, such as D1228, Y1230, and G1163. These sites, and many of the individual mutations, have been noted in prior reports, such as: D1228N/H/V/Y, Y1230C/H/N/S, G1163R (*Fernandes et al., 2021*; *Yao et al., 2023*; *Bahcall et al., 2022*; *Recondo et al., 2020a*; *Rotow et al., 2020*; *Fujino et al., 2019*; *Lu et al., 2017*; *Pecci et al., 2024*). Beyond these well-characterized sites, regions with sensitivity occurred throughout the kinase, primarily in loop-regions which have the greatest mutational tolerance in DMSO, but do not provide a growth advantage in the presence of an inhibitor.

In mapping positions with resistance to the crizotinib-bound kinase domain crystal structure (PDB 2WGJ), our DMS results further emphasize the emergence of resistance mutations at the ATP-binding site and direct-protein drug interacting residues (*Figure 3B–D*). Mutations to the hinge position, Y1159, and C-spine residues, including M1211 and V1092, introduce charge or are predicted to change the conformation of the pocket to clash with crizotinib but not ATP (*Figure 3E and F*). Outside of direct drug-protein interactions, positions I1084, T1261, Y1093, and G1242 displayed the largest resistance signals (*Figure 3A–D*). Structurally, I1084 is located in β1 at the roof of the ATP-binding pocket, and a mutation to His clashes with crizotinib's hinge-binding and solvent-front moieties without interfering with bound-ATP (*Figure 3E and F*). Y1093 is also at the roof of the ATP-binding site, residing in β2 (*Figure 3B*). However, its structural influence on resistance is unobvious. In all rotameric states, the R-group of Y1093 points away from the catalytic site and does not clash with crizotinib. We speculate that Y1093 mutations may negatively impact crizotinib's stability in the catalytic site compared to ATP, as ATP's triphosphate group is stabilized by the P-loop. Therefore, comparing crizotinib to DMSO highlights both known hotspots and rarer sites like I1084 and Y1093, which may contribute to resistance through conformational changes rather than disrupting direct inhibitor-protein contacts. Individual inhibitor resistance landscapes also aid in identifying target residues for novel drug design by providing insights into mutability and known resistance cases. This enables the selection of vectors for chemical elaboration with a potential lower risk of resistance development. Sites with mutational

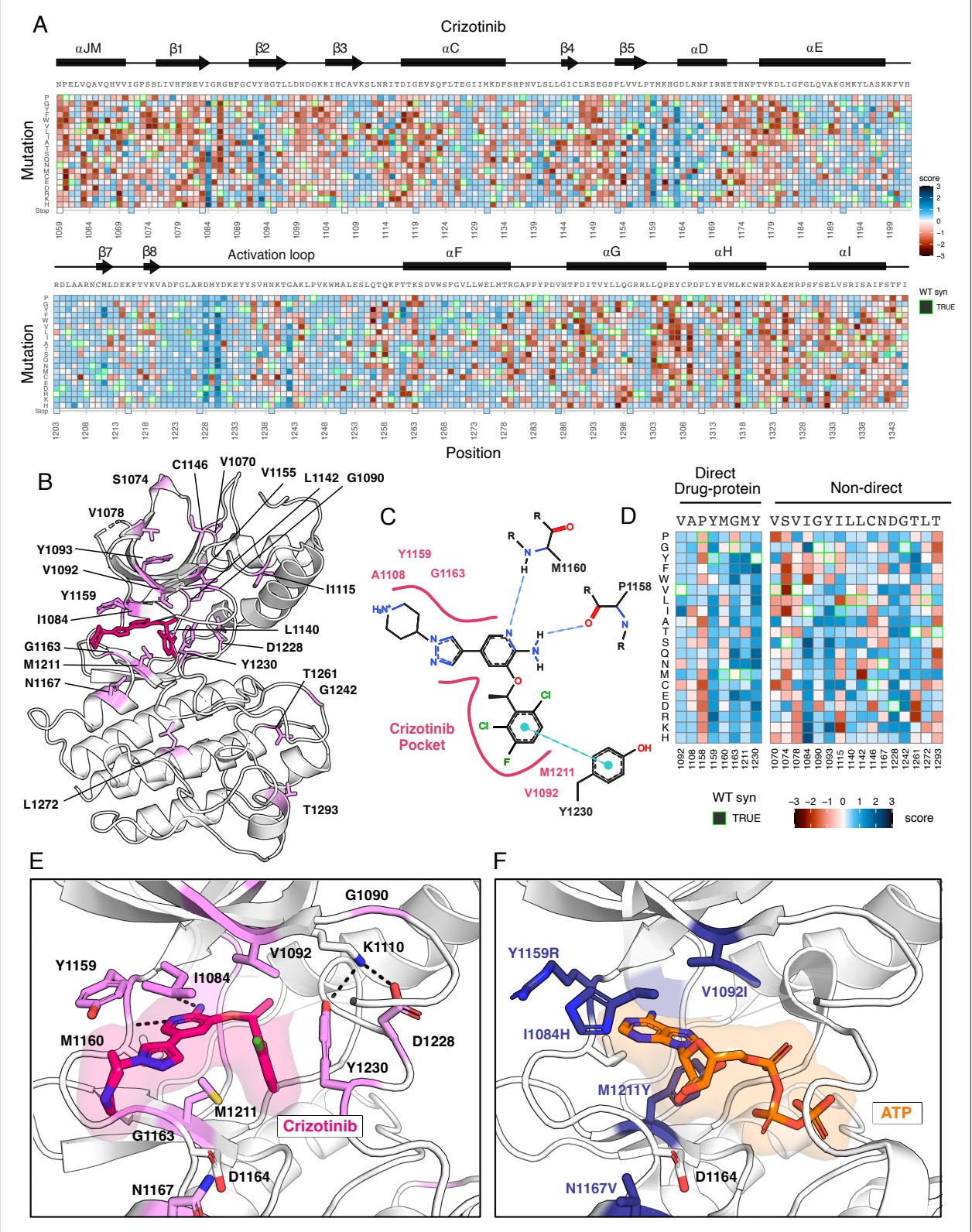

**Figure 3.** Novel resistance mutations identified and mapped for crizotinib. (**A**) Heatmap of crizotinib fitness scores subtracted from DMSO, scaled from loss-of-function (red) to gain-of-function (blue), with the wild-type protein sequence, secondary structure, kinase domain residue position, and mutational substitution annotated. Wild-type synonymous substitutions are outlined in green, and uncaptured mutations are in light yellow. (**B**) Resistance positions mapped onto the crizotinib-bound, MET crystal structure (PDB 2WGJ). Positions that contain one or multiple resistance

*Figure 3 continued on next page*

*Figure 3 continued*

mutations are labeled and colors are scaled according to the average score for the resistance mutations at each site. (**C**) 2D protein-drug interactions between crizotinib and the MET kinase domain (PDB 2WGJ) with pocket residues and polar and pi interactions annotated. Schematic generated through PoseEdit (**Diedrich et al., 2023**; https://proteins.plus/). (**D**) Condensed crizotinib heatmap displaying direct drug-protein interacting and non-direct resistance position. Again, fitness scores are scaled from loss-of-function (red) to gain-of-function (blue), wild-type synonymous substitutions are outlined in green, and uncaptured mutations are in light yellow. (**E**) Crizotinib binding site and pocket residues displayed with resistance positions highlighted (pink) and the wild-type residue and inhibitor interactions shown (PDB 2WGJ). (**F**) Resistance mutations modeled for I1084H, V1092I, Y1159R, M1211Y, and N1167 relative to ATP (PDB 3DKC).

profiles such as R1086 and C1091, located in the common drug target P-loop of MET, could be likely candidates for crizotinib.

## Resistance mutations identified for type I, type II, and type I ½ inhibitors

To assess the agreement between our DMS and previously annotated resistance mutations, we compiled a list of reported resistance mutations from recent clinical and experimental studies (**Pecci et al., 2024**; **Yao et al., 2023**; **Bahcall et al., 2022**; **Recondo et al., 2020b**; **Rotow et al., 2020**; **Fujino et al., 2019**; **Figure 4A and B**). Overall, previously discovered mutations are strongly shifted to a GOF distribution for the drugs where resistance is reported from treatment or experiment; in contrast, the distribution is centered around neutral for those sites for other drugs not reported in the literature (**Figure 4C**). However, even in cases such as L1195V, we observe GOF DMS scores indicative of resistance to previously reported inhibitors. Given this overall strong concordance with prior literature and clinical results, we can also provide hypotheses to clarify the role of mutations that are observed in combination with others. For example, H1094Y is a reported driver mutation that has been linked to resistance in METΔEx14 for glesatinib with either the secondary L1195V mutation or in isolation (**Recondo et al., 2020a**). However, in our assay H1094Y demonstrated slight sensitivity to gelesatinib, suggesting that either resistance is linked to the exon14 deletion isoform, the L1195V mutation, or a cellular factor not modeled well by the BaF3 system.

With this validation, we next wanted to identify the strongest potential resistance mutations. We identified unique resistance mutations enriched at the ATP-binding site across all inhibitors, yet also noticed discernible differences between type I and II inhibitors, the R-spine, and αC-helix (**Figure 5A–G**). Mapping inhibitor-specific positions and mutational scores, not only provides a mutation-level breakdown of inhibitor contributions to common resistance mutations, but also demonstrates differences in structural resistance enrichment across specific inhibitors (**Figure 5A–G**). To summarize this information, we next examined trends by grouping inhibitors by type.

Next, we filtered resistance mutations by their score and test statistics (**Figure 6—figure supplement 1**) and collapsed the information by inhibitor type, plotting the total frequency of resistance mutations at each position (**Figure 6A**). In this condensed heatmap, several common resistance positions emerged within and across inhibitor types to provide a broad view of 'hotspots'. Two positions stood out with the highest frequency of resistance: G1163 and D1228 (**Figure 6A–D**). Both sites are unsurprising due to their inhibitor interactions - G1163 is at the solvent front entrance of the active site and D1228 stabilizes an inactive conformation of the A-loop with an inhibitor bound (**Cui, 2014**; **Recondo et al., 2020a**; **Fernandes et al., 2021**). Located at the base of the active site, M1211 is a previously documented resistance site (**Tiedt et al., 2011**) and a C-spine residue (**Estevam et al., 2024**), which harbors a smaller number of resistance mutations for all inhibitor types within our analysis (**Figure 6A**). In contrast to these universal sites, Y1230 was a hotspot for type I and I ½ inhibitors, but not a major resistance site for type II inhibitors (**Figure 6A–D**). This specificity can be rationalized based on the role of Y1230 in stabilizing inhibitors through pi-stacking interactions (**Cui, 2014**). In contrast, F1200 and L1195 (**Bahcall et al., 2022**; **Recondo et al., 2020b**), are both hotspots for type II but not type I inhibitors (**Figure 6A–C**). Again, this effect can be rationalized structurally: both residues make direct contact with type II inhibitors, but not type I inhibitors.

Across all inhibitor types, there were a total of 17 shared variants with G1163, D1228, and M1211 being the most common (**Figure 6G**). The overall spatial pattern of mutations for each inhibitor type follows general principles that are expected based on their interactions. For example, I1084 is enriched as a resistance site for type I inhibitors, consistent with previous studies in hereditary papillary renal

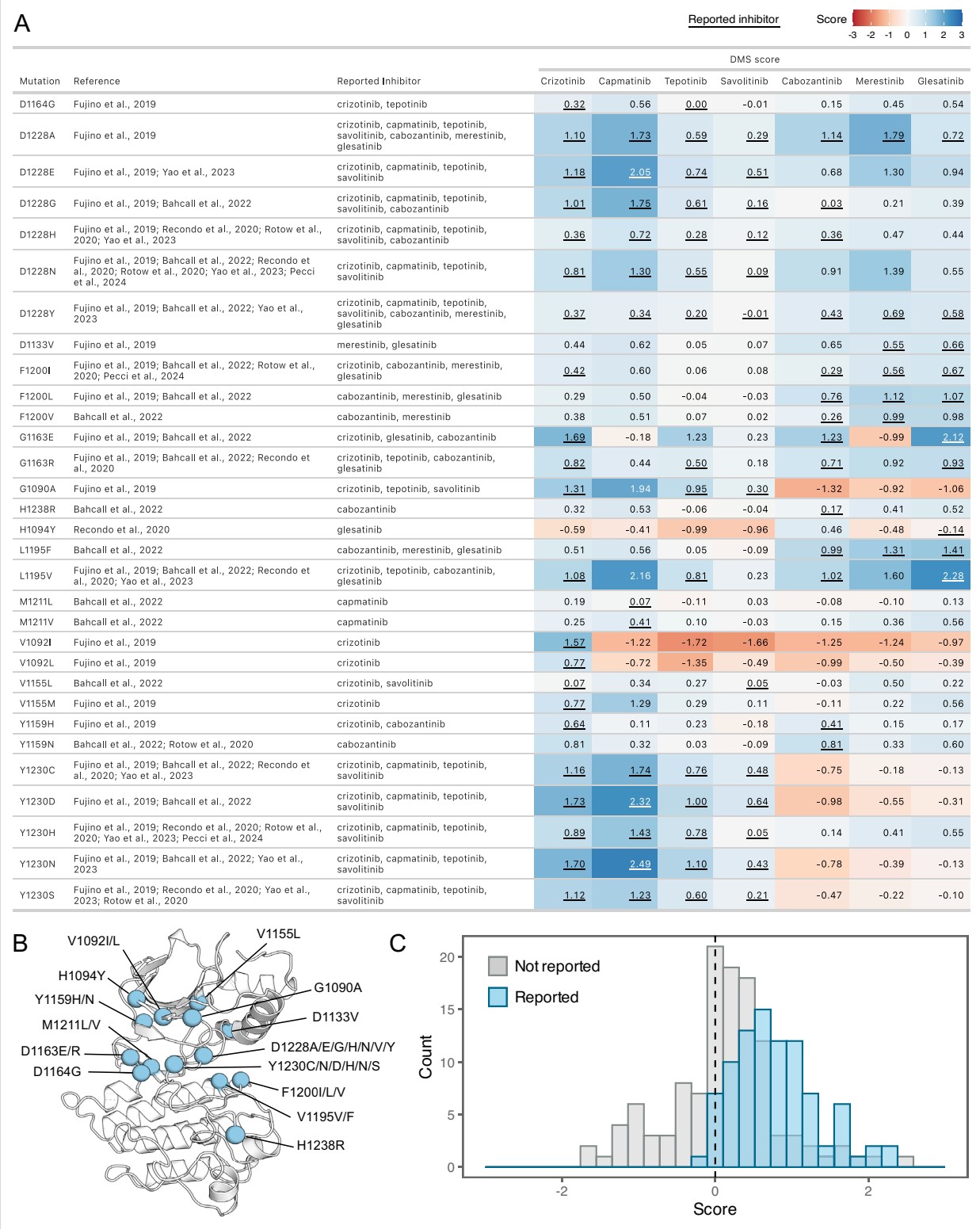

**Figure 4.** Comparison of previously reported resistance mutations with DMS fitness scores. (**A**) Data table summarizing reported resistance mutations from clinical and experimental studies. Inhibitors linked to reported resistance cases are listed, along with corresponding DMSO subtracted fitness scores from the DMS. The scores are represented with a color gradient ranging from loss-of-function (red) to gain-of-function (blue), with the reported inhibitor scores underlined. (**B**) Residue locations of previously reported resistance mutations mapped on a representative crystal structure as blue spheres (2WGJ). (**C**) Histograms of fitness scores from the DMS for previously annotated resistance mutations, comparing their reported inhibitor (blue) to non-reported inhibitor scores (gray).

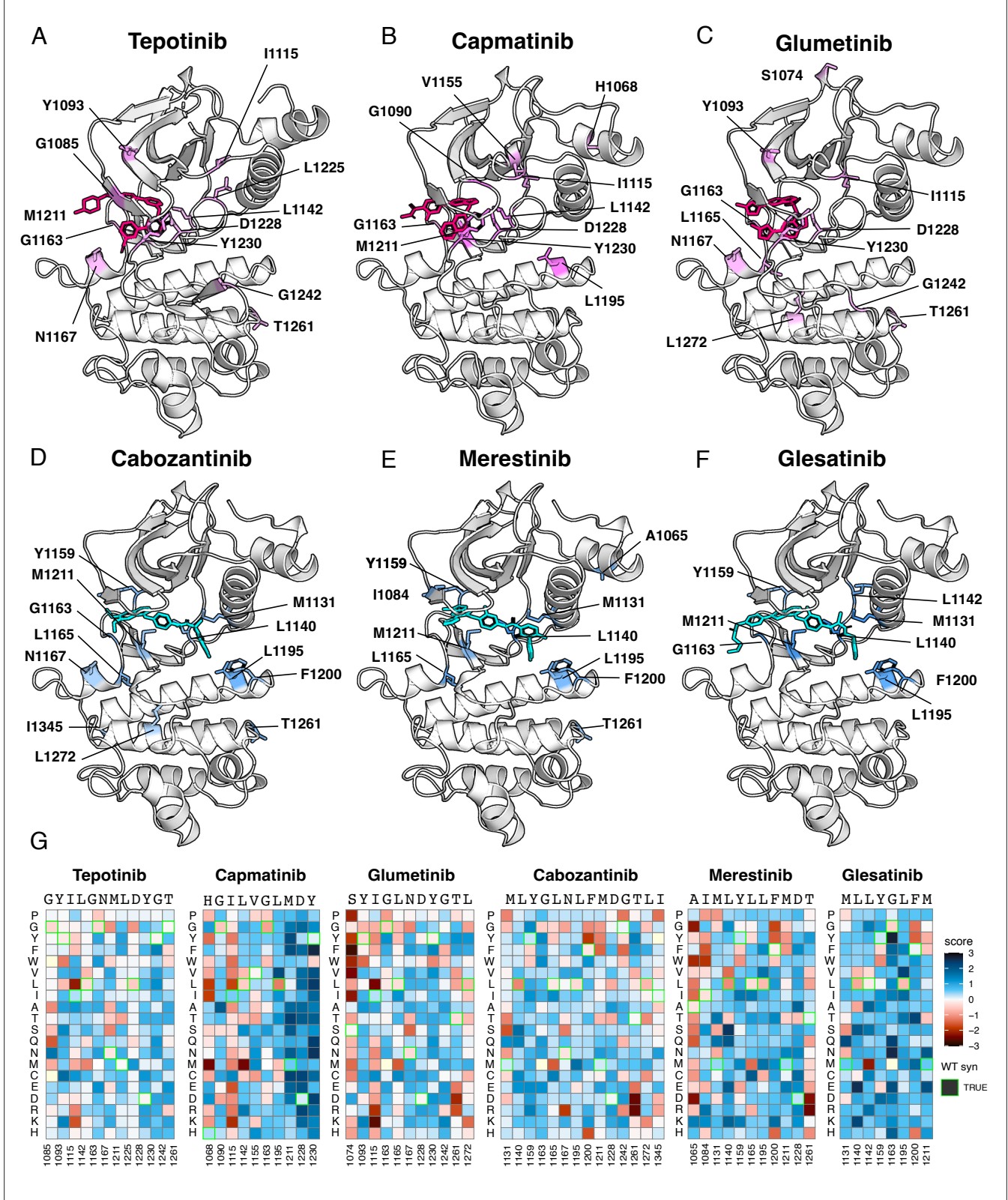

**Figure 5.** Resistance mutations mapped onto experimental and docked kinase domain structures for type I and type II inhibitors. (**A–F**) Resistance positions and average resistance mutational score mapped onto representative crystal structures (tepotinib, 4R1V; merestinib, 4EEV) and labeled (type I, pink; type II, blue). Inhibitors lacking experimental structures (capmatinib, cabozantinib, glumetinib, and glesatinib) were docked onto a representative type I (2WGJ) and type II (4EEV) crystal structure through AutoDock Vina (*Eberhardt et al., 2021*; *Trott and Olson, 2010*). (**G**) Heatmaps of each

*Figure 5 continued on next page*

*Figure 5 continued*

resistance position within an inhibitor DMS. Fitness scores are scaled from loss-of-function (red) to gain-of-function (blue). Wild-type synonymous substitutions are outlined in green, and mutations uncaptured by the screen are in light yellow.

cell carcinomas (*Guérin et al., 2023*). I1084 is located at the solvent front of the phosphate-loop (P-loop) of the kinase N-lobe (*Figure 1A*), which is responsible for stabilization of the ATP phosphate groups. This region of the kinase is leveraged for interactions with type I, but not type II inhibitors. In contrast, L1142, an R-spine residue, and L1140, which sits at the back of the ATP-binding pocket, are enriched for type II inhibitor resistance, consistent with their spatial locations (*Figure 6E–F*). Resistance mutations tend to cluster around the catalytic site across all types, but the shared mutations across different inhibitor types did not display a unifying pattern that evokes a simple rule for combining or sequencing inhibitors to counter resistance (*Figure 6H*).

## Differential sensitivities of the MET kinase domain to type I and type II inhibitors

Strategies aimed at preventing resistance, such as sequential or combination dosing of type I and type II inhibitors, have been explored and offer promise in preventing resistance (*Recondo et al., 2018*; *Bahcall et al., 2022*; *Fernandes et al., 2021*). However, the efficacy of these strategies is limited to the emergence of secondary resistance mutations, and specific inhibitor pairings are further limited to case examples of disparate effects. Using our DMS datasets, we sought to identify inhibitor pairings with the largest divergence in cross-sensitivity. By comparing the fitness landscape of each type I inhibitor to each type II inhibitor, we could again assess inhibitor response likeness based on correlations (*Figure 2A*; *Figure 7—figure supplement 1*). Type I and type II pairs with the highest correlations included capmatinib and glesatinib analog (*r*=0.92), suggesting a large overlapping fitness profile, in contrast to pairs with the lower correlations, like savolitinib and merestinib (*r*=0.7; *Figure 7A*; *Figure 7—figure supplement 1*). Overall, cabozantinib maintained the lowest average correlation with all type I inhibitors, making it the most divergent type II inhibitor within our screen, and potentially offering the least overlap in resistance (*Figure 7A*; *Figure 7—figure supplement 1*).

To narrow our characterization of cross-sensitivity, we focused on the inhibitor pair crizotinib and cabozantinib (*Figure 7B*; *Figure 7—figure supplement 1*). By statistically filtering mutations that are categorized as gain-of-function in one inhibitor, but loss-of-function in the other, a set of 44 mutations were identified as having crizotinib resistance and cabozantinib sensitivity, and 3 mutations with the opposite profile (*Figure 7B and C*). Structural mapping of divergent mutations further revealed enrichment at the N-lobe and typical protein-drug interaction sites like Y1230, G1163, and M1211 (*Figure 7B and C*). While these positions have precedence for resistance, as previously noted, they are also resistance hotspots across all inhibitor types (*Figure 6A*), where even mutations with differential sensitivities may be insufficient targets to counteract the reemergence of resistance, thus limiting the interchangeability of drugs.

Understanding which mutations have resistance profiles for only type I or type II inhibitors provides better leverage for sequential and combination dosing. To identify such mutations across our dataset, we further filtered variants that met our resistance metrics and were only observed for inhibitors of the same type. In again comparing crizotinib to cabozantinib, Y1093K was a mutation with one of the largest differences between crizotinib and cabozantinib, having a gain-of-function profile for crizotinib and loss-of-function for cabozantinib (*Figure 7B*). Interestingly, Y1093 is located in β2 of the N-lobe, at the roof of the ATP-binding site, and does not directly engage with crizotinib. We speculate this mutation potentially contributes to resistance by perturbing the packing of β1-β2 and altering the conformation of the ATP binding site in a manner that destabilizes crizotinib binding. When comparing the dose-response of Y1093K to the wild-type TPR-MET kinase domain, Y1093K shows a nearly 10-fold shift in crizotinib sensitivity with no difference in cabozantinib sensitivity (*Figure 7D*). In identifying mutations with the opposite profiles, resistance to cabozantinib and sensitivity to crizotinib, L1195M displayed the greatest differential scores (*Figure 7B*). L1195 is an αE-helix position with previously recorded resistance (L1195V/F), which our analysis further supports as a type II-only resistance hotspot (*Figure 6A*). Structurally, mutations like Met or Phe at 1195 clash with the fluorophenyl moiety of cabozantinib used to access and stabilize a deep, back pocket of the kinase in an inactive conformation, unlike crizotinib which occupies the solvent front and adenosine binding region of the ATP

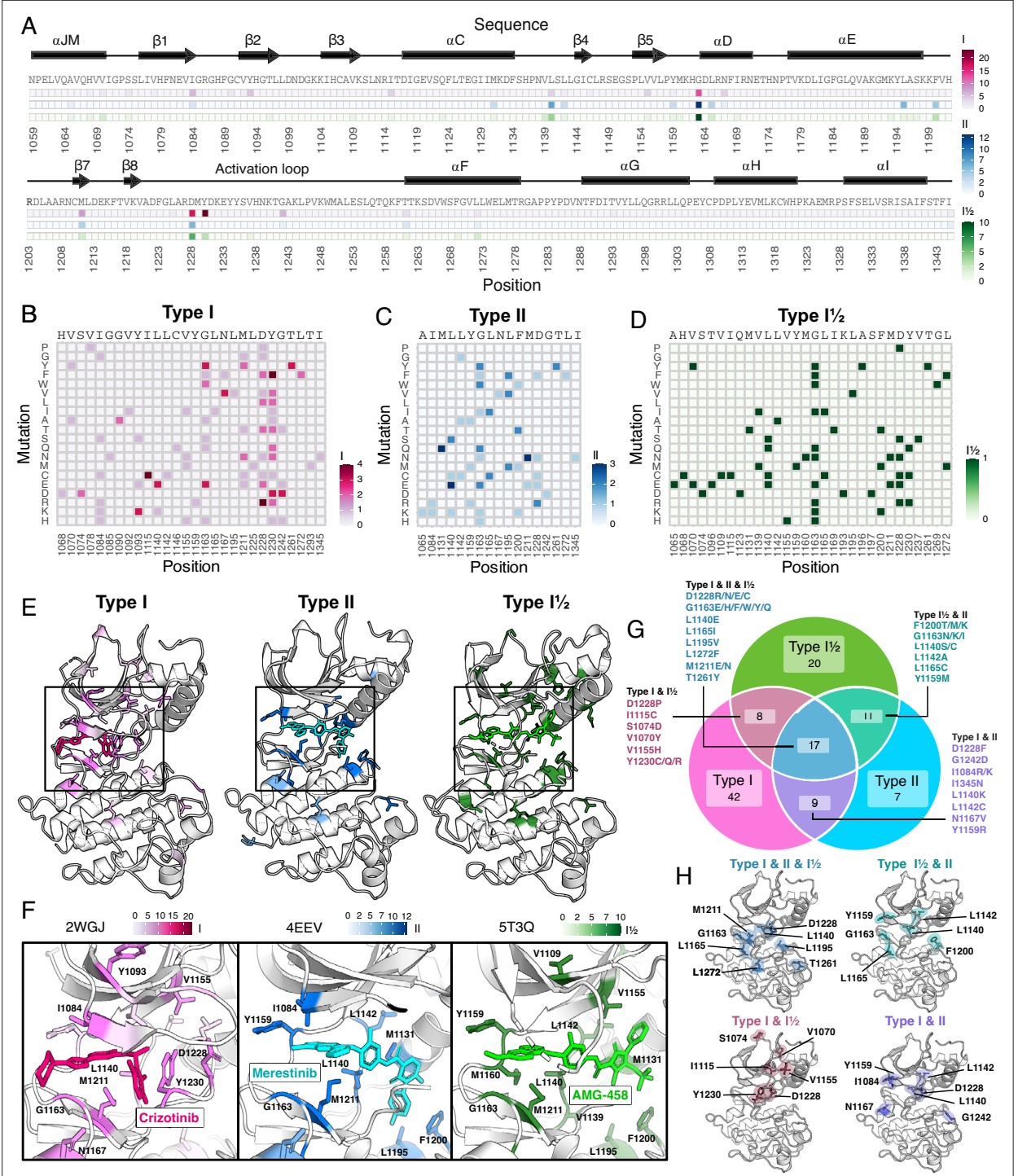

**Figure 6.** Resistance mutations and 'hotspots' identified for MET inhibitor types. (**A**) Collapsed heatmap of common resistance positions along the kinase domain, with the wild-type protein sequence and secondary structure annotated. Each tile represents a sum of counts for statistically filtered resistance mutations across all inhibitors for type I (pink), type II (blue), and the type I½ inhibitor AMG-458 (green), with the scale bar reflecting counts of resistance mutations across respective inhibitor types. (**B–D**) Expanded heatmap showing each resistance position and the counts for each specific resistance mutation across all inhibitor types type I (pink), type II (blue), and the type I½ inhibitor AMG-458 (green). Wild-type sequence and variant change are annotated. (**E–F**) Average frequency of resistance mutations for each mapped on to a representative type I (crizotinib, 2WGJ) and type II (merestinib, 4EEV) crystal structure, alongside the type I½, AMG-458 structure (5T3Q), with associated scale bars. Individual positions with high resistance mutation frequencies are annotated on each structure, with a zoom-in of the bound inhibitor and surrounding resistance sites. (**G**) Venn

*Figure 6 continued on next page*

*Figure 6 continued*

diagram showing mutations shared among type I (pink), type II (light blue), and type I½ (green). (**H**) Structurally mapped (PDB 2WGJ) resistance positions shared among type I, II, I½ (blue-gray), type I and II (purple), type I and I½ (dusty rose), type II and I½ (teal) inhibitors.

The online version of this article includes the following figure supplement(s) for figure 6:

**Figure supplement 1.** Statistically filtered resistance mutations for grouped type I, type II, and type I½ inhibitors for MET.

binding site. In comparing the dose-response of L1195M to the wild-type TPR-MET kinase domain, we find that L1195M is refractory to all concentrations of cabozantinib tested, but still sensitive to crizotinib (*Figure 7D*). Beyond a type I and type II pairing, such cross-resistance identification can be further applied to identify differential sensitivities within an inhibitor group (*Figure 7—figure supplement 2*), which can further expand opportunities for inhibitor-specific sensitivity in therapy and drug design.

## Identification of biophysical contributors to inhibitor-specific fitness landscapes using machine learning

Machine learning models originally developed for predicting protein structure *Jumper et al., 2021*; *Rives et al., 2021*; *Lin et al., 2023* have been adapted for predicting protein-ligand complexes

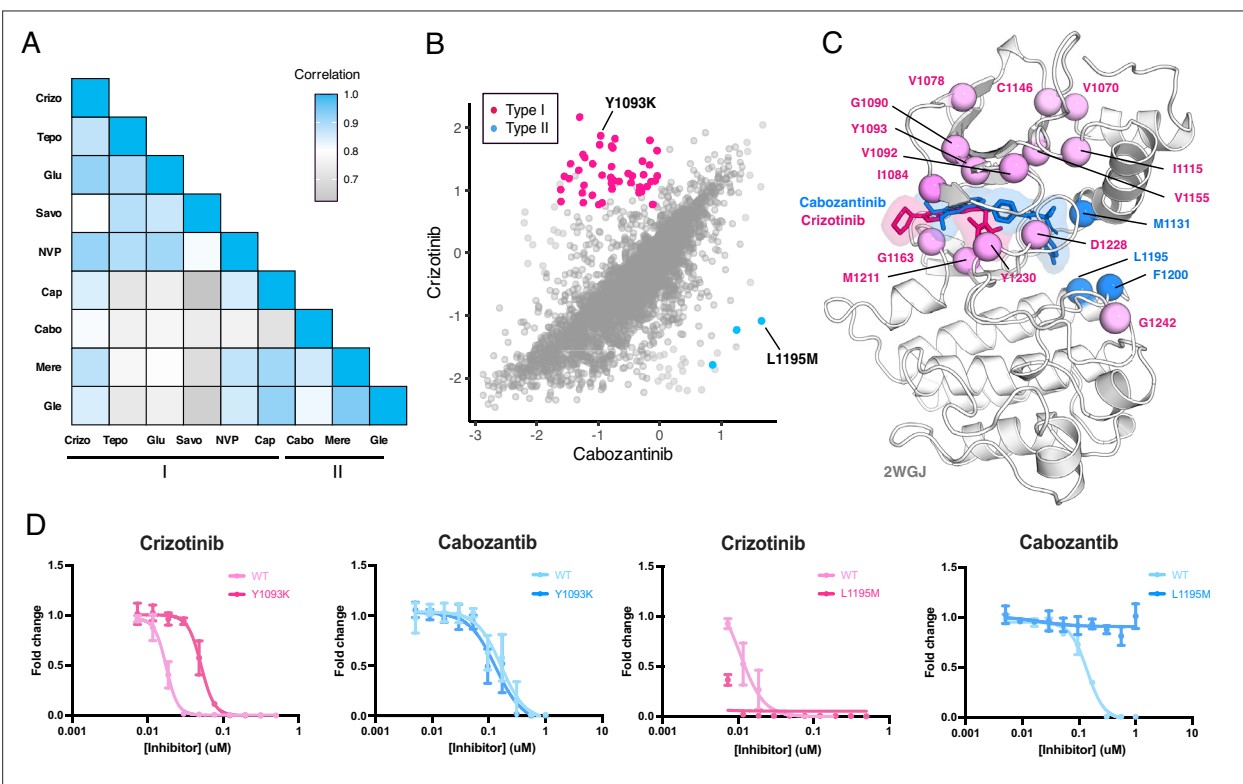

**Figure 7.** MET kinase domain differential sensitivities revealed for type I and type II inhibitors. (**A**) Heatmap showing Pearson correlation values for all combinations of screened type I and type II inhibitors. Correlations were determined from DMSO subtracted fitness scores (**B**) Correlation plot correlation plot of DMSO subtracted fitness scores for crizotinib and cabozantinib. Mutations with differential scores are highlighted for type I (pink) and type II (blue). (**C**) Average scores of mutations with differential sensitivities within inhibitor pairs mapped and annotated in respective crystal structures (crizotinib, 2WGJ; cabozantinib, docked into 4EEV). Positions that are gain-of-function for type I but loss-of-function in type II are highlighted in pink, whereas positions that are gain-of-function for type II but loss-of-function in type I are highlighted in blue. (**D**) Dose-response curves for crizotinib and cabozantinib in Ba/F3 cells expressing TPR-MET (full MET intracellular domain) harboring mutations at Y1093K and L1195M. Dose-response for each inhibitor concentration is represented as the fraction of viable cells relative to the TKI free control.

The online version of this article includes the following figure supplement(s) for figure 7:

**Figure supplement 1.** Cross-comparison of type I and type II inhibitor pairs.

**Figure supplement 2.** Cross-comparison analysis of inhibitors within the same type.

(*Bryant et al., 2023*), and predicting fitness values from DMS studies (*Meier et al., 2021*; *Brandes et al., 2023*; *Jones et al., 2020*). In particular, protein language models have shown the ability to estimate the functional effects of sequence variants in correlation with DMS data (*Rives et al., 2021*; *Meier et al., 2021*; *Brandes et al., 2023*). We observed that ESM-1b, a protein language model (*Rives et al., 2021*), predicts the fitness of variants in the untreated/DMSO condition (correlation 0.50) much better than it does for the inhibitor treated datasets (correlation 0.28). This difference in predictive ability is likely because the language model is trained on sequences in the evolutionary record and fitness in the presence of inhibitors does not reflect a pressure that has operated on evolutionary timescales.

To overcome this limitation and improve the predictive properties of the ESM approach, we sought to augment the model with additional features that reflect the interactions between protein and inhibitors that are not present in the evolutionary record (*Chen and Guestrin, 2016*). While our features can account for some changes in MET-mutant conformation and altered inhibitor binding pose, the prediction of these aspects can likely be improved with new methods. There are several challenges associated with this task, including the narrow sequence space explored, high correlations between datasets, and the limited chemical space explored by the 11 inhibitors. We used an XGBoost regressor framework and designed a test-train-validation strategy to account for these issues (*Figure 8A*), exploring many features representing conformation, stability, inhibitor-mutation distance, and inhibitor chemical information (*Figure 8—figure supplement 1*). To avoid overfitting, we introduced several constraints on the monotonicity and the precision of certain features. The final model uses a subset of the features we tested and improves the performance from 0.28 to 0.37 (*Figure 8B and C*). The model primarily improves the correlation by shifting the distribution of predicted fitness values to center around drug sensitivity, reflecting the pressures that are not accounted for by ESM-1b (*Figure 8D*). Nonetheless, many resistant mutations are correctly predicted by the new model.

To examine whether the model could help interpret the mechanisms of specific mutations, we examined several cases with notable improved predictions as the model increased in complexity (*Figure 8E and F*). For some mutations, as in Y1230D, we observe a gradual improvement in prediction for each set of features, suggesting that resistance relies on multiple factors. For other mutations, such as N1167K, we see a single set of features driving the improvement, which suggests much more dominant driving forces. Lastly, in other mutations, like G1290D, the models trained with different features can over or under predict the true value, demonstrating the value of combining features together. The reliance on simple features helps identify some of the major factors in drug resistance and sensitization such as distance to the inhibitor and active/inactive conformation; however, improved feature engineering and coverage of both sequence and chemical space will likely be needed to create a more interpretable model.

## Discussion

Tyrosine kinase inhibitors have revolutionized the treatment of many diseases, but the development of resistance creates a significant challenge for long term efficacy. Many strategies, including sequential dosing (*Attwood et al., 2021*; *Recondo et al., 2020a*), are being explored to overcome resistance. Our DMS of the MET receptor tyrosine kinase domain, performed against a panel of varying inhibitors offers a framework for experimentally identifying resistance and sensitizing mutations in an activated kinase context for different inhibitors. By massively screening the effect of a nearly comprehensive library of amino acid mutations in the MET kinase domain against 11 inhibitors, some generalizable patterns emerged. In concordance with the binding mode of both type I and II inhibitors, residues that commonly confer resistance, or act as 'hotspots', were mapped to previously reported sites like D1228, Y1230, M1211, G1163 (*Figure 4*), and novel sites like I1084, L1140, L1142, T1261, and L1272 (*Figure 5*). Annotation of hotspots also offers an opportunity to inform inhibitor selection based on likelihood of cross-inhibitor resistance (*Figure 6*). For instance, I1084 is a hotspot for type I and II inhibitors within our study that displayed wild-type sensitivity to the type I½ inhibitor screened (*Figure 6*). Understanding positions with high resistance frequencies that are distal from the ATP-binding site also offers a design opportunity for allosteric inhibitors that can target cancer-associated and resistance-associated regions within the N- and C-lobe (*Mingione et al., 2023*).

Nevertheless, similar to its ability in identifying resistance for inhibitors, our parallel DMS also demonstrated the ability to detect non-selective drugs, with the example of tivantinib. Despite being

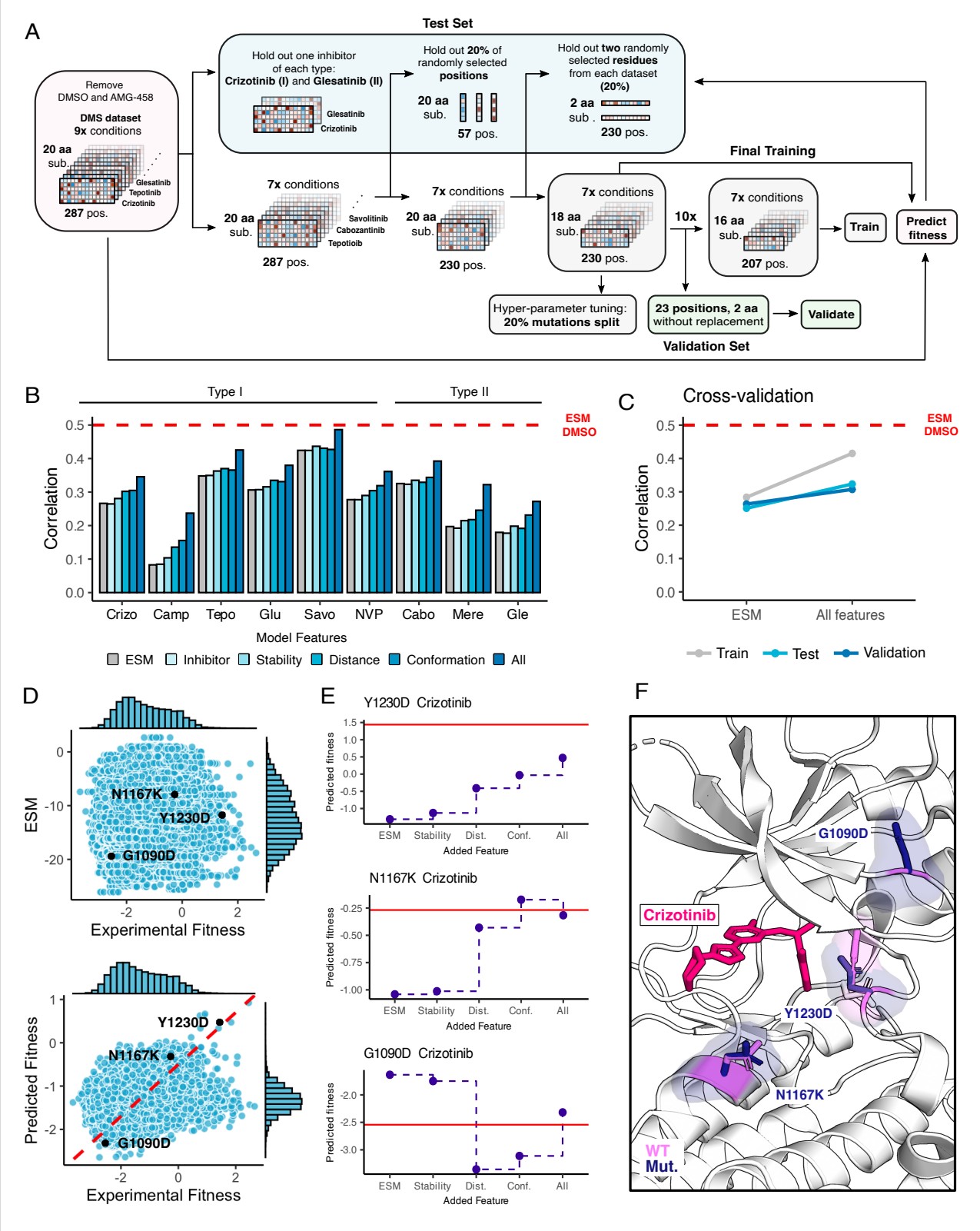

**Figure 8.** Inhibitor-bound variant fitness predicted from a machine learning model trained on the MET DMS dataset. (**A**) Model architecture outlining the information flow and inputs for model training, validation, fitness predictions, and prediction tests. (**B**) Improvement in correlation between experimental and predicted fitness for each inhibitor with usage of different kinds of features. (**C**) Cross-validation trends between the baseline ESM model and the model with all features incorporated. (**D**) Scatter plots of predictions versus experimental fitness scores of the baseline ESM model (top)

*Figure 8 continued on next page*

*Figure 8 continued*

compared to the model with all features (bottom), with a dashed cross-graph line in red displayed. (**E**) Residue-level analysis of feature significance in fitness predictions (ESM, stability, distance, conformation, all features). The Rosace experimental score is shown as a red line. (**F**) Residues with improved predictions mapped on a crizotinib-bound MET kinase domain (PDB 2WGJ). Predicted resistance mutations (dark purple) modeled relative to the wild-type residue (pink).

The online version of this article includes the following figure supplement(s) for figure 8:

**Figure supplement 1.** Distribution and visualization of features used in the XGBoost machine learning models.

a proposed MET-selective inhibitor, like several others, tivantinib failed clinical trials, and follow-up studies suggested cytotoxicity and off-target binding as the culprit (*Michieli and Di Nicolantonio, 2013*; *Basilico et al., 2013*; *Katayama et al., 2013*; *Fujino et al., 2019*) - a scenario that is not uncommon to antitumor drugs that do not advance to the clinic (*Lin et al., 2019*). To this effect, the ability of DMS to differentiate between selective compounds provides a unique prospect for developing inverse structure-activity relationships, whereby varying protein sequence both inhibitor specificity and resistance can be learned.

Reported cancer mutations in databases such as OncoKB or cBioPortal are useful for patient data and cancer type reporting (*Suehnholz et al., 2024*; *Chakravarty et al., 2017*; *Cerami et al., 2012*). A recent analysis of these databases aided in annotation of mutations observed within patient populations (*Pecci et al., 2024*). This study used pre-clinical models to examine a subset of these mutations and identified sensitivities to multiple inhibitors and confirmed clinical responses of two rare driver mutations (H1094Y and F1200I) to elzovantinib, a type Ib inhibitor (*Pecci et al., 2024*). Their results are consistent with the predictions of our DMS, illustrating the potential value of having a broad dictionary of inhibitor sensitivity and resistance patterns.

Finally, a significant challenge of inhibitor screening is the considerable time and cost involved, even at high-throughput. While docking has accelerated the prioritization of compounds for protein targeting and screening in silico (*Sadybekov and Katritch, 2023*), prediction of drug resistance is of high interest in informing iterative drug design. Screening for resistance in the early stages of drug design is particularly useful for obtaining inhibitors that can be effective in the long-term by optimizing protein-inhibitor interactions in the wildtype and functionally silent mutant context (*Pisa and Kapoor, 2020*). While base-editor approaches can rapidly screen for inhibitor resistance mutations within full-length, endogenous genes, undersampling of rare variants due to lower coverage is a significant caveat (*Dorighi et al., 2024*), compared to DMS where nearly full coverage is achieved and controlled. A full landscape of mutational effects can help to predict drug response and guide small molecule design to counteract acquired resistance. The ability to define molecular mechanisms towards that goal will likely require more purposefully chosen chemical inhibitors and combinatorial mutational libraries to be maximally informative. The ideas motivating our ML-model, which combines protein language models and biophysical/chemical features, to novel inhibitors could eventually be used to profile resistance and sensitivity for novel and unscreened small molecules, greatly extending the scale of kinase inhibitor repositioning for second-line therapies.

# Materials and methods
## Mammalian cell culturing
Ba/F3 cells (DSMZ) were maintained and passaged in 90% RPMI (Gibco), 10% HI-FBS (Gibco), 1% penicillin/streptomycin (Gibco), and 10 ng/ml IL-3 (Fisher), and incubated at 37 °C with 5% $CO_2$. Cells were passaged at or below 1.0e6 cells/ml to avoid acquired IL-3 resistance, and regularly checked for IL-3 dependence by performing 3 x PBS (Gibco) washes and outgrowth in the absence of IL-3.

Plat-E cells stably expressing retroviral envelope and packaging plasmids were originally gifted by Dr. Wendell Lim, and maintained in 90% DMEM, HEPES (Gibco), 10% HI-FBS (Gibco), 1% penicillin/streptomycin (Gibco), 10 µg/ml blasticidin, 1 µg/ml puromycin. Cells were cultured at 37 °C with 5% $CO_2$ and maintained under blasticidin and puromycin antibiotic pressure unless being transfected.

## Dose response and IC50 determination of inhibitors
Unless otherwise stated, all inhibitors used in this study were purchased from SelleckChem.

Ba/F3 cells stably expressing TPR-MET and TPR-METΔEx14 were washed with DPBS (Gibco) 3 x times to remove IL-3, puromycin, penicillin, and streptomycin. Cells were resuspended in 90% RPMI and 10% FBS, and were seeded in the wells of a 96-well, round-bottom plate at a density of 2.5e4 cells/ml in 200 μl. Cells were incubated for 24 hr to allow kinase-driven signaling. The next day, inhibitors were added to triplicate rows of cells at a concentration range of 0–10 μM (twofold dilutions), and allowed to incubate for 72 hr post TKI addition. CellTiter-Glo reagent (Promega) was mixed at a 1:1 ratio with cells to lyse and provide a luminescence readout, which was measured on a Veritas luminometer. Cell numbers were determined from a Ba/F3 cell and ATP standard curve generated according to the manufacturer's instructions. Dose response curves were fitted using GraphPad Prism with the log(inhibitor) vs. response, variable slope function. Data are presented as cell viability normalized to the fold change from the TKI free control.

## MET kinase domain variant library generation, cloning, and library introduction into Ba/F3

In this study, we repurposed cell lines transduced with TPR-MET and TPR-METΔEx14 kinase domain variant libraries, previously reported in *Estevam et al., 2024*. All libraries were generated, transfected, and tested in parallel.

In short, the MET kinase domain sequence used in this study spans amino acid positions 1059–1345, which includes the full kinase domain (aa 1071–1345) and a small region of the juxtamembrane (aa 1059–1070). The variant DNA library was synthesized by Twist Bioscience, containing one mammalian high-usage codon per amino acid. A 'fill-in' library was generated to introduce an early stop control codon every 11 amino acids evenly spaced across the sequence. In addition, mutations at positions with failed synthesis (positions 1194 and 1278) were generated and added at equimolar concentration into the variant library. The kinase domain variant library was introduced into two different cloning backbones, one carrying the TPR-fusion sequence with the wild-type juxtamembrane sequence (aa 963–1058), wild-type C-terminal tail (aa 1346–1390), and IRES-EGFP (pUC19_kozak-TPR-METΔEx14-IRES-EGFP) and the other carrying the TPR-fusion sequence with an exon 14 skipped juxtamembrane sequence (aa 1010–1058), wild-type C-terminal tail (aa 1346–1390), and IRES-mCherry (pUC19_kozak-TPR-MET-IRES-mCherry). The libraries were transformed in MegaX 10 beta cells (Invitrogen), propagated in 50 mL LB and Carbinacillin at 37 °C to an OD of 0.5, and then midiprepped (Zymo). Library coverage was determined by colony count of serial dilutions from the recovery plate at varying dilutions (1:100, 1:1 k, 1:10 k, 1:100 k, 1:1 M).

The full TPR-METΔEx14-IRES-EGFP and TPR-MET-IRES-mCherry variant libraries were then shuttled into the mammalian retroviral MSCV backbone (addgene) through restriction enzyme digest with MluI-HF (NEB) and MfeI-HF (NEB), then ligated into the empty backbone with T4 ligase (NEB). Ligations were DNA cleaned (Zymo), electroporated into ElectroMAX Stbl4 Competent Cells (Thermo Fisher), plated on LB-agar bioassay plates with Carbenicillin, incubated at 37 °C, then colonies were scraped into 50 mL LB and midi-prepped for transfections (Zymo).

Variant libraries were transfected into Plat-E cells for retroviral packaging using Lipofectamine3000 (Invitrogen) following the manufacturer's for a T-175 scale, and using a total of 46 μg DNA. 48 hr post-transfection, the viral supernatant was harvested, passed through a 0.45 μm sterile filter, then concentrated with Retro-X concentrator (TakaraBio) using a 1:4 ratio of concentrator to supernatant. The concentrated virus was titered in Ba/F3 to determine the proper volume for a transduction MOI of 0.1–0.3. The viral titer was calculated from the percent of fluorescent cells and viral dilution. To generate the DMS transduced cell lines, 6 million cells were spinfected at an MOI of 0.1, in triplicate. Then infected cells were selected with 1 μg/ml puromycin in 4 days, with fluorescence and cell counts tracked each day.

## DMS time point selection and sample preparation

All screening conditions were performed and handled in parallel for TPR-MET and TPR-METΔEx14 libraries across all independent conditions and biological replicates.

For each biological replicate, a stock of 4.0e6 cells transduced with TPR-MET and TPR-METΔEx14 kinase domain variants was thawed and expanded for 48 hr in the presence of IL-3 and puromycin to prevent pre-TKI selection to reach a density for screen seeding. Each batch of cells were grown to a density of 72 million cells to be split into 12 dishes (15 cm) for each selection condition. Cells were

first washed with DPBS (Gibco) three times to remove IL-3 and antibiotics. Cells were resuspended in 90% RPMI and 10% FBS, counted, and split across 12 dishes (15 cm) at a density of 6 million cells in 30 mL. A total of 6 million cells from each replicate was harvested and pelleted at 250 x $g$ to serve as the 'time point 0' pre-selection sample (T0).

To begin selection of each replicate for each library, DMSO was added to the control plate (0.01% final) while the appropriate IC50 concentration of inhibitor was added to each respective plate (independent pool of cells). Three time points post T0 were collected for each library replicate and inhibitor condition for a total of 4 time points (T0, T1, T2, T3). Time points were harvested every two doublings (~72 hr) across 12 days; 6 million cells were harvested for each condition and pelleted at 250 x $g$ for 5 min; 2.0e5 cells/ml were split at every time point and maintained either in DMSO or TKI at the appropriate concentration to maintain cellular growth rates under inhibitor selection.

The gDNA of each time point sample was isolated with the TakaraBio NucleoSpin Blood Quick-Pure kit the same day the cells were harvested. gDNA was eluted in 50 µl of elution buffer provided by the kit, using the high concentration and high yield elution manufacturer's protocol. Immediately after gDNA was isolated, 5 µg of gDNA was used for PCR amplification of the target MET KD gene to achieve the proper variant coverage. A 150 µl PCR master mix was prepared for each sample using the TakaraBio PrimeStar GXL system according to the following recipe: 30 µl 5 X PrimeStar GXL buffer, 4.5 µl 10 µM forward primer (0.3 µM final), 4.5 µl 10 µM reverse primer (0.3 µM final), 5 µg gDNA, 12 µl 10 mM dNTPs (2.5 mM each NTP), 6 µl GXL polymerase, nuclease free water to a final reaction volume of 150 µL. The PCR master mix for each sample was split into three PCR tubes with 50 µl volumes for each condition and amplified with the following thermocycler parameters: initial denaturation at 98 °C for 30 s, followed by 24 x cycles of denaturation at 98 °C for 10 s, annealing at 60 °C for 15 s, extension at 68 °C for 14 s, and a final extension at 68 °C for 1 min.

PCR samples were stored at –20 °C until all time points and replicates were harvested and amplified, so as to prepare all final samples for NGS together with the same handling and sequence them in the same pool to prevent sequencing bias.

## Library preparation and next-generation sequencing

After all time points were selected, harvested, and PCR amplified, the target gene amplicon was isolated from gDNA by gel purification (Zymo), for a total of 222 samples. The entire 150 µl PCR reaction for each sample was mixed with 1 X NEB Purple Loading Dye (6 X stock) and run on a 0.8% agarose, 1 X TBE gel, at 100 mA until there was clear ladder separation and distinct amplicon bands. The target amplicons were gel excised and purified with the Zymo Gel DNA Recovery kit. To remove excess agarose contamination, each sample was then further cleaned using the Zymo DNA Clean and Concentrator-5 kit and eluted in nuclease free water. Amplicon DNA concentrations were then determined by Qubit dsDNA HS assay (Invitrogen).

Libraries were then prepared for deep sequencing using the Nextera XT DNA Library Prep kit in a 96-well plate format (Illumina). Manufacturer's instructions were followed for each step: tagmentation, indexing and amplification, and clean up. Libraries were indexed using the IDT for Nextera Unique Dual Indexes Set A,B and C (Illumina). Then, indexed libraries were quantified using the Agilent TapeStation with HS D5000 screen tape (Agilent) and reagents (Agilent). DNA concentrations were further confirmed with a Qubit dsDNA HS assay (Invitrogen). All samples were manually normalized and pooled at 10 nM (MET and METΔEx14 in the same pool). The library was then paired-end sequenced (SP300) on two lanes of a NovaSeq6000.

## MET kinase domain variant analysis and scoring
### Enrich2 scoring
Our approach followed the one used for our initial MET DMS experiments (*Estevam et al., 2024*). Sequencing files were obtained from the sequencing core as demultiplexed fastq.gz files. The reads were first filtered for contamination and adapters using BBDuk, then the paired reads were error-corrected and merged with BBMerge and mapped to the reference sequence using BBMap (all from BBTools; *Bushnell, 2015*). Read consequences were determined and counted using the AnalyzeSaturationMutagenesis tool in GATK v4 (*van der Auwera and O'Connor, 2020*). This is further processed by use of a script to filter out any variants that are not expected to be in the library (i.e. variants due to errors in sequencing, amplification, etc). The final processed count files were then analyzed with

Enrich2 (*Rubin et al., 2017*), using weighted least squares and normalizing to wildtype sequences (NCBI SRA BioProject PRJNA1136906).

## Rosace scoring

We used Rosace to analyze experiments of different conditions (DMSO or inhibitors) independently. In order to make the scores more comparable and interpretable across conditions, we modified the original Rosace software so that the output scores reflect the scale of cell doubling rate between every contiguous time point. For example, in an ideal experiment, if the wild-type cell doubling rate is 2 and its score is 0 by wild-type normalization, a score of –2 means that the cells are not growing ($2^{(2-2)}$) and a score of 1 means that the cells are doubling three times (increasing to $2^{(2+1)}$ times the original count) between every contiguous time point.

The input to Rosace is the filtered count files provided by the AnalyzeSaturationMutagenesis tool described in the above section. From here, we filtered variants by the mean count ($\geq 4$) and the proportion of 0 count across replicates and time points ($\leq 10/12$) and before inhibitor selection at T0 ($\leq 2/3$). Second, we normalized the counts using wild-type normalization with log2 transformation rather than the default natural log transformation to maintain the doubling rate scale. Finally, the normalized count is regressed on time intervals (t = {1, 2, 3, 4}) instead of the entire time span (t = {1/4, 2/4, 3/4, 4/4}) so that the resulting score reflects the growth rate between every contiguous time point.

## Statistical filtering and resistance classification

In mathematical terms, we define the raw Rosace fitness scores of a mutation in DMSO as $\beta_{v,DMSO}$ and in a specific inhibitor condition as $\beta_{v,inh}$. The scores of wildtype variants were normalized to 0, and we denote them as $\beta_{wt,DMSO} = 0$ and $\beta_{wt,inh} = 0$. Growth rate of wild-type cells under different inhibitor selections were controlled to be identical (two doublings between every time point), so even though raw Rosace scores are computed independently per condition, $\beta_{v,inh}$ are directly comparable between inhibitor conditions.

Within one condition (DMSO or inhibitor), according to convention, we call variants with $\beta_{v,inh} \gg 0$ or $\beta_{v,DMSO} \gg 0$ 'gain-of-function' and $\beta_{v,inh} \ll 0$ or $\beta_{v,DMSO} \ll 0$ 'loss-of-function'. With scores from multiple conditions, we presented three types of filtering strategies and produced the following classification: inhibitor-specific 'resistance mutation', inhibitor-specific 'resistance position', and 'loss-of-function' and 'gain-of-function' mutation in the context of growth rate differential with and without an inhibitor.

We stress the different interpretations of 'gain-of-function' and 'loss-of-function' labels. Within one condition, this label is a general term to describe whether the function of protein is perturbed by the mutation. In contrast, the latter describes the difference with and without a given inhibitor, canceling effects of folding, expression, and stability and targeting only the inhibitor sensitivity function of the protein.

A 'resistance mutation' is specific to a certain inhibitor, and it satisfies the chained inequality $\beta_{v,inh} \gg \beta_{wt,inh} = \beta_{wt,DMSO} \geq \beta_{v,DMSO}$. The first inequality specifies that the growth rate of a resistant mutation needs to be much larger than that of the wild-type in the presence of the inhibitor, and we used the one-sided statistical test $\beta_{v,inh} > 0.5$ with the test statistics cutoff 0.1. The second inequality specifies that in DMSO, the growth rate of that mutation is equal to or lower than that of the wild-type, ensuring that the resistance behavior we see is specific to that inhibitor, not that it grows faster under every condition, and thus we used the effect size cutoff $\beta_{v,DMSO} \leq 0$.

A 'resistance position' is a position that contains at least one 'resistance mutation' to a certain inhibitor.

In the context of growth rate differential with and without an inhibitor, a mutation is 'gain-of-function' if it has a higher growth rate in the presence of the inhibitor than in its absence, which is one feature of 'resistance mutation'. It is 'loss-of-function' if it grows faster in the absence of the inhibitor. To label the mutations accordingly, we first computed a recentered Rosace score for each mutation under inhibitor selection $\gamma_{v,inh} = \beta_{v,inh} - \beta_{v,DMSO}$, and define 'gain-of-function' $\gamma_{v,inh} > 0.75$ and 'loss-of-function' $\gamma_{v,inh} < 0$ in the differential sensitivity analysis.

## Machine learning modeling

### Feature selection for the machine learning model

Interpretable features of the MET sequence variants and inhibitors were carefully chosen to be incrementally added to a model. To extract structural features from inhibitor bound mutant complexes, we used Umol to predict the structures of all the MET kinase variants bound to each of the inhibitors (*Bryant et al., 2023*). The input to Umol is the MET kinase variant sequence, SMILES string of the inhibitor and list of residues lining the putative binding pocket. The predicted complexes (MET kinase bound to inhibitor) were relaxed using OpenMM (*Eastman et al., 2013*). To ensure the inhibitor in the predicted structures are in the same pose as compared to reference structures, we tethered the predicted inhibitor structure to the reference pose using a modified version of the script available in https://github.com/Discngine/rdkit_tethered_minimization, copy archived at *Discngine, 2019*. The reference pose for crizotinib, NVP-BVU972, Merestenib and Savolitinib were taken from the corresponding crystal structures in the PDB - 2WGJ, 3QTI, 4EEV and 6SDE, while the reference pose for cabozantinib, capmantinib, glumetinib, and glesatinib analog were taken from the structures docked using Autodock Vina (see Kinase domain structural analysis). Following this, the tethered inhibitors were redocked back to the predicted variant structures using Autodock Vina (*Eberhardt et al., 2021*). We also extracted features from wild-type MET kinase structures. The features could be broadly classified into four categories: inhibitor, stability, distance, conformation and inhibitor binding. Apart from these, ESM Log Likelihood Ratio was used as a feature in all models that we trained. Each of the feature categories that we explored and the rationale behind choosing them are explained below:

### ESM Log Likelihood Ratio (ESM LLR)

ESM1b is an unsupervised protein language model trained on a large set of protein sequences from UniProt that has successfully learned protein fitness patterns (*Rives et al., 2021*; *Lin et al., 2023*). By including a mask token at a given position in the sequence, the log-likelihoods of all amino acid substitutions can be extracted from the model. The ratio between ESM1b log-likelihoods for the mutant and wildtype amino acids provides a score that indicates the fitness of each variant in the mutational scan, with log-likelihood ratios having precedent as a variant predictor (*Rives et al., 2021*; *Lin et al., 2023*). The predictions used here were obtained using esm-variants webserver (https://huggingface.co/spaces/ntranoslab/esm_variants) (*Brandes et al., 2023*).

### Inhibitor features

- Inhibitor molecular weight: We calculated the molecular weight of each inhibitor as a feature.
- Ligand RMSD: We structurally superposed the predicted variant structure onto the corresponding wildtype structure and calculated the RMSD between the predicted, re-docked inhibitor and the reference inhibitor structure (*Figure 8—figure supplement 1J*)

### Stability features

- $\Delta\Delta\Delta G$ and $\Delta G$: Because inhibitor types are largely distinguished based on binding configuration, we reasoned that the difference in stability contributed by each mutation between given binding states (e.g. Type I bound state vs. a Type II bound state) could contribute to the success of the predictor. To compute the stability difference, we used structural representatives for type-I bound (2WGJ) and type II bound (4EEV) MET kinase and calculated the change in free energy ($\Delta\Delta G$) of every possible mutation at every position using ThermoMPNN (*Dieckhaus et al., 2023*). The difference in $\Delta\Delta G$ between type-I bound and type-II bound structures ($\Delta\Delta\Delta G$) for every variant was added as a feature to the XGBoost model to capture the difference in stabilization from the mutation in the Type I or Type II bound state (*Figure 8—figure supplement 1C*). We also used the predicted $\Delta$ score of the corresponding inhibitor type-bound structure directly as a feature. For instance, if the input data corresponds to a mutation to Alanine at position 1065 in the presence of glumetinib (a type I inhibitor), the difference between $\Delta$ predicted for the 1065 A variant for the type-I bound (2WGJ) and for type II bound (4EEV) structure is used as a feature ($\Delta$). The $\Delta$ predicted for the 1065 A variant for the type-I bound (2WGJ) structure is also used as a feature.

## Distance features

- Residue to ATP distance: Proximity to the ATP-binding site indicates the ability of the given residue to influence inhibitor binding given that Type I and Type II inhibitors are ATP competitive. To include this feature, the distance between C-alpha residue atoms and the centroid of bound ATP in a representative structure (PDB 3DKC) was calculated and the distance corresponding to each position was added as a feature (*Figure 8—figure supplement 1D*).
- Inhibitor distance: This is the shortest distance between the inhibitor and mutated residue in the predicted variant-inhibitor complexes. (*Figure 8—figure supplement 1G*).

## Conformational features

- MET crystal structure RMSF: The extent of flexibility at the mutation position could be significantly affected by the mutation, which in turn can affect the function of the variant. To account for this, we utilized the structural information abundantly available for MET kinases in PDB. We structurally aligned all crystal structures of human MET kinases with resolution better than 3 Å (81 structures) using mTM-align (*Dong et al., 2018*) and calculated the Root Mean Squared Fluctuation at every residue position using Prody (*Zhang et al., 2021*; *Figure 8—figure supplement 1F*).
- Residue RMSD: We structurally superposed the predicted variant structure onto the corresponding wildtype structure and calculated the RMSD between the mutant and wildtype residue at the mutation position (*Figure 8—figure supplement 1I*)

## Inhibitor binding features

- RF-Score: To quantify the binding strength between the inhibitor and the variant protein structure, we calculated the RF-score, which is a random forest-based approach to predict protein-ligand binding affinity (*Wójcikowski et al., 2017*)
- Pocket volume, hydrophobicity score, and polarity score: Changes to the binding pocket in terms of volume and hydrophobicity due to mutations could affect the interaction and binding between the inhibitor and variant. These effects were brought in as features into the model by calculating the binding pocket volume, hydrophobicity score, and polarity score of the binding pocket using fpocket (*Le Guilloux et al., 2009*; *Figure 8—figure supplement 1H*).

This category of features are not part of the best performing model shown in *Figure 8*.

Apart from these categories, we calculated the difference in volume between the wildtype and mutated residue at a given position and added it as a feature (**Δ**) since residue volume changes upon mutation could contribute to steric hindrance (*Figure 8—figure supplement 1E*). This feature is also not part of the best performing model.

This led to a total of 14 interpretable features to evaluate our models on. We trained and tested a total of 8192 models by considering all possible numbers and combinations of these features (keeping ESM LLR as a constant feature in all models). The hyperparameter tuning, cross-validation, training and testing of each of these models are described in detail below.

## Training and selecting the predictive model

An XGBoost regressor model, which is a gradient boosting method based on decision trees as the base learner (*Chen and Guestrin, 2016*), was used to predict DMS fitness scores in presence of inhibitors. Given the relatively small dataset we are using here, the models are prone to overfitting. Hence, we used monotonic constraints on features that had a monotonic relationship with the experimental fitness scores. ESM LLR score and Δ have a positive and negative correlation with the experimental fitness scores respectively (*Figure 8—figure supplement 1A*). Therefore, ESM LLR was constrained positively and Δ was constrained negatively by assigning 1 and –1 respectively to the 'monotone_constraints' parameter in Python XGBoost. This ensures that the monotonic relationship between the input feature and the target value is maintained during predictions.To further prevent overfitting, we binned the values of the 12 remaining into four or five bins and assigned the median of the bin as their value. The bins were chosen such that one or two bins would contain the majority of feature values. The distribution of these twelve features are shown in *Figure 8—figure supplement 1B*. The bins of

each feature are shown as red dashed lines on the histograms. Model performance was evaluated using Pearson's R and mean squared error (MSE).

Experimental fitness scores of MET variants in the presence of DMSO and AMG458 were ignored in model training and testing since having just one set of data for a type I ½ inhibitor and DMSO leads to learning by simply memorizing the inhibitor type, without generalizability. The remaining dataset was split into training and test sets to further avoid overfitting (*Figure 8A*). The following data points were held out for testing - (a) all mutations in the presence of one type I (crizotinib) and one type II (glesatinib analog) inhibitor, (b) 20% of randomly chosen positions (columns) and (c) all mutations in two randomly selected amino acids (rows; e.g. all mutations to Phe, Ser). After splitting the dataset into train and test sets, the train set was used for XGBoost hyperparameter tuning and cross-validation. For tuning the hyperparameters of each of the XGBoost models, we held out 20% of randomly sampled data points in the training set and used the remaining 80% data for Bayesian hyperparameter optimization of the models with Optuna (*Akiba et al., 2019*), with an objective to minimize the mean squared error between the fitness predictions on 20% held out split and the corresponding experimental fitness scores. The following hyperparameters were sampled and tuned: type of booster (booster - gbtree or dart), maximum tree depth (max_depth), number of trees (n_estimators), learning rate (eta), minimum leaf split loss (gamma), subsample ratio of columns when constructing each tree (colsample_bytree), L1 and L2 regularization terms (alpha and beta) and tree growth policy (grow_policy - depthwise or lossguide). After identifying the best combination of hyperparameters for each of the models, we performed 10-fold cross validation (with re-sampling) of the models on the full training set. The training set consists of data points corresponding to 230 positions and 18 amino acids. We split these into 10 parts such that each part corresponds to data from 23 positions and 2 amino acids. Then, at each of 10 iterations of cross-validation, models were trained on 9 of 10 parts (207 positions and 16 amino acids) and evaluated on the 1 held out part (23 positions and 2 amino acids). Through this protocol we ensure that we evaluate performance of the models with different subsets of positions and amino acids. The average Pearson correlation and mean squared error of the models from these 10 iterations were calculated and the best performing model out of 8192 models was chosen as the one with the highest cross-validation correlation. The final XGBoost models were obtained by training on the full training set and also used to obtain the fitness score predictions for the validation and test sets. These predictions were used to calculate the inhibitor-wise correlations shown in *Figure 8B*.

## Kinase domain structural analysis

Unless otherwise stated, all structural analysis was performed on PyMOL. Structural mapping incorporated tools from the Bio3D bioinformatics package in R (*Grant et al., 2006*). Inhibitors that lacked an experimental crystal structure were docked into a representative type I (2WGJ) or type II (4EEV) structure with AutoDock Vina (*Eberhardt et al., 2021*). Existing ligands in both the structures were removed in silico and the proteins prepared for docking using AutoDockTools by adding polar hydrogens and Kollman charges. The inhibitors were also prepared using AutoDockTools by adding polar hydrogens and charges and identifying rotatable torsions. A grid box which dictates the search space for the docking tool was defined approximately around the region where the existing ligands in 2WGJ and 4EEV were bound. The energy range and exhaustiveness of docking was set to 3 and 8, respectively. AutoDock Vina was made to output 5 modes for each ligand. Capmatenib and glumetinib (type I inhibitors) were docked on to 2WGJ and glesatinib analog and cabozantinib (type II inhibitors) were docked on to 4EEV.

## Acknowledgements

Sequencing was performed at the UCSF CAT, supported by UCSF PBBR, RRP IMIA, and NIH 1S10OD028511-01 grants. This work was supported by NIH CA239604 to EAC, NJ, JSF; HHMI Hanna Gray Fellowship and UCSF QBI Fellow program to WCM; NIH R01LM013434 to JAC; NIH GM145238 and the UCSF Program for Breakthrough Biomedical Research, funded in part by the Sandler Foundation, to JSF.

# Additional information

## Competing interests

Eric A Collisson: Consultant at IHP Therapeutics, Valar Labs, Tatara Therapeutics and Pear Diagnostics, reports receiving commercial research grants from Pfizer, and has stock ownership in Tatara Therapeutics, HDT Bio, Clara Health, Aqtual, and Guardant Health. Natalia Jura: A founder of Rezo Therapeutics and a shareholder of Rezo Therapeutics, Sudo Therapeutics, and type6 Therapeutic; is a SAB member of Sudo Therapeutics, type6 Therapeutic and NIBR Oncology; the Jura laboratory has received sponsored research support from Genentech, Rezo Therapeutics and type6 Therapeutics. James S Fraser: A consultant for, has equity in, and receives research support from Relay Therapeutics and is a consultant for Octant Bio. The other authors declare that no competing interests exist.

## Funding

| Funder | Grant reference number | Author |
| --- | --- | --- |
| National Cancer Institute | CA239604 | Eric A Collisson<br>Natalia Jura<br>James S Fraser |
| Howard Hughes Medical Institute | | Willow Coyote-Maestas |
| National Institutes of Health | LM013434 | John A Capra |
| National Institute of General Medical Sciences | GM145238 | James S Fraser |

The funders had no role in study design, data collection and interpretation, or the decision to submit the work for publication.

## Author contributions

Gabriella O Estevam, Conceptualization, Data curation, Formal analysis, Validation, Investigation, Visualization, Methodology, Writing – original draft, Writing – review and editing; Edmond Linossi, Formal analysis, Methodology, Writing – review and editing; Jingyou Rao, Software, Formal analysis, Writing – review and editing; Christian B Macdonald, Ashraya Ravikumar, Karson M Chrispens, Data curation, Formal analysis, Writing – review and editing; John A Capra, Conceptualization, Project administration, Writing – review and editing; Willow Coyote-Maestas, Conceptualization, Supervision, Project administration, Writing – review and editing; Harold Pimentel, Conceptualization, Data curation, Supervision, Project administration, Writing – review and editing; Eric A Collisson, Conceptualization, Supervision, Funding acquisition, Project administration, Writing – review and editing; Natalia Jura, Conceptualization, Funding acquisition, Project administration, Writing – review and editing; James S Fraser, Conceptualization, Funding acquisition, Writing – original draft, Project administration, Writing – review and editing

## Author ORCIDs

Gabriella O Estevam ⓘ https://orcid.org/0000-0002-9142-7805
Edmond Linossi ⓘ https://orcid.org/0000-0002-8039-573X
Christian B Macdonald ⓘ https://orcid.org/0000-0002-0201-8832
John A Capra ⓘ https://orcid.org/0000-0001-9743-1795
Willow Coyote-Maestas ⓘ https://orcid.org/0000-0001-9614-5340
Eric A Collisson ⓘ https://orcid.org/0000-0001-8037-9388
Natalia Jura ⓘ https://orcid.org/0000-0001-5129-641X
James S Fraser ⓘ https://orcid.org/0000-0002-5080-2859

Reviewer #2 (Public review): https://doi.org/10.7554/eLife.101882.3.sa1
Reviewer #3 (Public review): https://doi.org/10.7554/eLife.101882.3.sa2
Author response https://doi.org/10.7554/eLife.101882.3.sa3

## Additional files

**Supplementary files**
MDAR checklist

**Data availability**
The sequencing data has been deposited at the NCBI SRA (BioProject PRJNA1136906). Original data files, analysis, and source code is available at https://github.com/fraser-lab/MET_kinase_Inhibitor_DMS (copy archived at *Estevam, 2024*).

The following dataset was generated:

| Author(s) | Year | Dataset title | Dataset URL | Database and Identifier |
|---|---|---|---|---|
| Estevam et al. | 2024 | Inhibitor-based deep mutational scanning of MET kinase | https://www.ncbi.nlm.nih.gov/bioproject/?term=PRJNA1136906 | NCBI BioProject, PRJNA1136906 |

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
