## [Editor Report · eLife Assessment]

This manuscript provides an **important** overview of potential resistance mutations within MET Receptor Tyrosine Kinase. The evidence supporting the findings is **convincing** - it should be pointed out that the approach is comparatively new for the application of protein kinases and the results are therefore of potentially great value. The results will be of value for clinicians facing drug resistance mutations, computational biologists who are training models of drug resistance mechanisms and biologists with an interest in cell signaling.

---

## [Referee Report · Reviewer #2 (Public review)]

Summary:

This manuscript provides a comprehensive overview of potential resistance mutations within MET Receptor Tyrosine Kinase and defines how specific mutations affect different inhibitors and modes of target engagement. The goal is to identify inhibitor combinations with the lowest overlap in their sensitivity to resistant mutations and determine if certain resistance mutations/mechanisms are more prevalent for specific modes of ATP-binding site engagement. To achieve this, the authors measured the ability of ~6000 single mutants of MET's kinase domain (in the context of a cytosolic TPR fusion) to drive IL-3-independent proliferation (used as a proxy for activity) of Ba/F3 cells (deep mutational profiling) in the presence of 11 different inhibitors. The authors then used co-crystal and docked structures of inhibitor-bound MET complexes to define the mechanistic basis of resistance and applied a protein language model to develop a predictive model of inhibitor sensitivity/resistance.

Strengths:

The major strengths of this manuscript are the comprehensive nature of the study and the rigorous methods used to measure the sensitivity of ~6000 MET mutants in a pooled format. The dataset generated will be a valuable resource for researchers interested in understanding kinase inhibitor sensitivity and, more broadly, small molecule ligand/protein interactions. The structural analyses are systematic and comprehensive, providing interesting insights into resistance mechanisms. Furthermore, the use of machine learning to define inhibitor-specific fitness landscapes is a valuable addition to the narrative. Although the ESM1b protein language model is only moderately successful in identifying the underlying mechanistic basis of resistance, the authors' attempt to integrate systematic sequence/function datasets with machine learning serves as a foundation for future efforts.

Weaknesses:

The main limitation of this study is that the authors' efforts to define general mechanisms between inhibitor classes were only moderately successful due to the challenge of uncoupling inhibitor-specific interaction effects from more general mechanisms related to the mode of ATP-binding site engagement. However, this is a minor limitation that only minimally detracts from the impressive overall scope of the study.

---

## [Referee Report · Reviewer #3 (Public review)]

Summary:

In the manuscript 'Mapping kinase domain resistance mechanisms for the MET receptor tyrosine kinase via deep mutational scanning' by Estevam et al, deep mutational scanning is used to assess the impact of ~5,764 mutants in the MET kinase domain on the binding of 11 inhibitors. Analyses were divided by individual inhibitor and kinase inhibitor subtype (I,II, I 1/2, and III). While a number of mutants were consistent with previous clinical reports, novel potential resistance mutants were also described. This study has implications for the development of combination therapies, namely which combination of inhibitors to avoid based on overlapping resistance mutant profiles. While one suggested pair of inhibitors with least overlapping resistance mutation profiles was suggested, this manuscript presents a proof of concept toward a more systematic approach for improved selection of combination therapeutics. Furthermore, in a final part of this manuscript the data was used to train a machine learning model, the ESM-1b protein language model augmented with an XG Boost Regressor framework, and found that they could improve predictions of resistance mutations above the initial ESM-1b model.

Strengths:

Overall this paper is a tour-de-force of data collection and analysis to establish a more systematic approach for the design of combination therapies, especially in targeting MET and other kinases, a family of proteins significant to therapeutic intervention for a variety of diseases. The presentation of the work is mostly concise and clear with thousands of data points presented neatly and clearly. The discovery of novel resistance mutants for individual MET inhibitors, kinase inhibitor subtypes within the context of MET, and all resistance mutants across inhibitor subtypes for MET has clinical relevance. However, probably the most promising outcome of this paper is the proposal of the inhibitor combination of Crizotinib and Cabozantib as Type I and Type II inhibitors, respectively, with the least overlapping resistance mutation profiles and therefore potentially the most successful combination therapy for MET. While this specific combination is not necessarily the point, it illustrates a compelling systematic approach for deciding how to proceed in developing combination therapy schedules for kinases. In an insightful final section of this paper, the authors approach using their data to train a machine learning model, perhaps understanding that performing these experiments for every kinase for every inhibitor could be prohibitive to applying this method in practice.

Weaknesses:

This paper presents a clear set of experiments with a compelling justification. The content of the paper is overall of high quality. Below are mostly regarding clarifications in presentation.

Two places could use more computational experiments and analysis, however. Both are presented as suggestions, but at least a discussion of these topics would improve the overall relevance of this work. In the first case it seems that while the analyses conducted on this dataset were chosen with care to be the most relevant to human health, further analyses of these results and their implications of our understanding of allosteric interactions and their effects on inhibitor binding would be a relevant addition. For example, for any given residue type found to be a resistance mutant are there consistent amino acid mutations to which a large or small or effect is found. For example is a mutation from alanine to phenylalanine always deleterious, though one can assume the exact location of a residue matters significantly. Some of this analysis is done in dividing resistance mutants by those that are near the inhibitor binding site and those that aren't, but more of these types of analyses could help the reader understand the large amount of data presented here. A mention at least of the existing literature in this area and the lack or presence of trends would be worthwhile. For example, is there any correlation with a simpler metric like the Grantham score to predict effects of mutations (in a way the ESM-1b model is a better version of this, so this is somewhat implicitly discussed).

Indeed, this discussion relates to the second point this manuscript could improve upon: the machine learning section. The main actionable item here is that this results section seems the least polished and could do a better job describing what was done. In the figure it looks like results for certain inhibitors were held out as test data - was this all mutants for a single inhibitor, or some other scheme? Overall I think the implications of this section could be fleshed out, potentially with more experiments. As mentioned in the 'Strengths' section, one of the appealing aspects of this paper is indeed its potential wide applicability across kinases -- could you use this ML model to predict resistance mutants for an entirely different kinase? This doesn't seem far-fetched, and would be an extremely compelling addition to this paper to prove the value of this approach.

Another area in which this paper could improve its clarity is in the description of caveats of the assay. The exact math used to define resistance mutants and its dependence on the DMSO control is interesting, it is worth discussing where the failure modes of this procedure might be. Could it be that the resistance mutants identified in this assay would differ significantly from those found in patients? That results here are consistent with those seen in the clinic is promising, but discrepancies could remain. Furthermore a more in depth discussion of the MetdelEx14 results is warranted. For example, why is the DMSO signature in Figure 1 - supplement 4 so different from that of Figure 1? And finally, there is a lot of emphasis put on the unexpected results of this assay for the tivantinib "type III" inhibitor - could this in fact be because the molecule "is highly selective for the inactive or unphosphorylated form of c-Met" according to Eathiraj et al JBC 2011? These points are addressed in previous work (Estevam et al 2024) or in the detailed methods section, but are not obvious in the main text of the paper.

This paper is crisply written with beautiful figures, and the complexity of the data is easy to understand from an in depth discussion of the mutants that have been previously reported.

Finally, the potential impacts and follow-ups of this excellent study could be used as a resource for the community both as a dataset and as a proof of concept. It is exciting that his approach can be altered and/or improved in the future to facilitate the general application of this approach for combination therapies and the understanding of mechanism for other targets.

Comments on revisions:

Thank you for your additions and changes - they have improved the quality of this paper.

---

## [Author Response]

The following is the authors’ response to the original reviews.

**Public Reviews:**

**Reviewer #1 (Public review):**
Summary:In this work, the authors present a cornucopia of data generated using deep mutational scanning (DMS) of variants in MET kinase, a protein target implicated in many different forms of cancer. The authors conducted a heroic amount of deep mutational scanning, using computational structural models to augment the interpretation of their DMS findings.Strengths:This powerful combination of computational models, experimental structures in the literature, dose-response curves, and DMS enables them to identify resistance and sensitizing mutations in the MET kinase domain, as well as consider inhibitors in the context of the clinically relevant exon-14 deletion. They then try to use the existing language model ESM1b augmented by an XGBoost regressor to identify key biophysical drivers of fitness. The authors provide an incredible study that has a treasure trove of data on a clinically relevant target that will appeal to many.

We thank Reviewer 1 for their generous assessment of our manuscript!

Weaknesses:However, the authors do not equally consider alternative possible mechanisms of resistance or sensitivity beyond the impact of mutation on binding, even though the measure used to discuss resistance and sensitivity is ultimately a resistance score derived from the increase or decrease of the presence of a variant during cell growth.

For this resistance screen, Ba/F3 was a carefully chosen cellular selection system due to its addiction to exogenously provided IL-3, undetected expression of endogenous RTKs (including MET), and dependence on kinase transgenes to promote signaling and growth under IL-3 withdrawal. Together this allows for the readout of variants that alter kinase-driven proliferation without the caveat of bypass resistance. In our previous phenotypic screen (Estevam et al., 2024, eLife), we also carefully examined the impact of all possible MET kinase domain mutations both in the presence and absence of IL-3 withdrawal, but no inhibitors. There, we identified a small group of mutations that were associated with gain-of-function behavior located at conserved regulatory motifs outside of the catalytic site, yet these mutations were largely sensitive to inhibitors within this screen.

Here, the majority of resistance mutations were located at or near the ATP-binding pocket, suggesting an impact on resistance through direct drug interactions. However, there was also a small population of distal mutations that met our statistical definitions of resistance. Within the crizotinib selection, sites such as T1293, L1272, T1261, amongst others, demonstrated resistance profiles but were located in C-lobe away from the catalytic site. While we did not experimentally validate these specific mutations, it is possible that non-direct drug binders instead promote resistance through allosteric or conformational mechanisms which preserve kinase activity and signaling. Indeed, our ML framework explicitly included conformational and stability effects as significant in improving predictions.

We would be happy to further discuss any specific alternative resistance mechanisms Reviewer 1 has in mind! Thank you for highlighting this!

There are also points of discussion and interpretation that rely heavily on docked models of kinase-inhibitor pairs without considering alternative binding modes or providing any validation of the docked pose. Lastly, the use of ESM1b is powerful but constrained heavily by the limited structural training data provided, which can lead to misleading interpretations without considering alternative conformations or poses.

The majority of our interpretations are grounded in the X-ray structures of WT MET bound to the inhibitors studied (or close analogs). The use of docked models (note - to mutant structures predicted by UMol, not ESM, that can have conformational changes) is primarily in the ML part of the manuscript. Indeed, in our models, conformational and binding mode changes are taken into account as features (see Ligand RMSD, Residue RMSD). There are certainly improved methods (AF3 variants) emerging that might have even more power to model these changes, but they come with greater computational costs and are something we will be evaluating in the future.

We added to the results section: “While our features can account for some changes in MET-mutant conformation and altered inhibitor binding pose, the prediction of these aspects can likely be improved with new methods.”

**Reviewer #2 (Public review):**
Summary:This manuscript provides a comprehensive overview of potential resistance mutations within MET Receptor Tyrosine Kinase and defines how specific mutations affect different inhibitors and modes of target engagement. The goal is to identify inhibitor combinations with the lowest overlap in their sensitivity to resistant mutations and determine if certain resistance mutations/mechanisms are more prevalent for specific modes of ATP-binding site engagement. To achieve this, the authors measured the ability of ~6000 single mutants of MET's kinase domain (in the context of a cytosolic TPR fusion) to drive IL-3-independent proliferation (used as a proxy for activity) of Ba/F3 cells (deep mutational profiling) in the presence of 11 different inhibitors. The authors then used co-crystal and docked structures of inhibitor-bound MET complexes to define the mechanistic basis of resistance and applied a protein language model to develop a predictive model of inhibitor sensitivity/resistance.Strengths:The major strengths of this manuscript are the comprehensive nature of the study and the rigorous methods used to measure the sensitivity of ~6000 MET mutants in a pooled format. The dataset generated will be a valuable resource for researchers interested in understanding kinase inhibitor sensitivity and, more broadly, small molecule ligand/protein interactions. The structural analyses are systematic and comprehensive, providing interesting insights into resistance mechanisms. Furthermore, the use of machine learning to define inhibitor-specific fitness landscapes is a valuable addition to the narrative. Although the ESM1b protein language model is only moderately successful in identifying the underlying mechanistic basis of resistance, the authors' attempt to integrate systematic sequence/function datasets with machine learning serves as a foundation for future efforts.

We thank Reviewer 2 for their thoughtful assessment of our manuscript!

Weaknesses:The main limitation of this study is that the authors' efforts to define general mechanisms between inhibitor classes were only moderately successful due to the challenge of uncoupling inhibitor-specific interaction effects from more general mechanisms related to the mode of ATP-binding site engagement. However, this is a minor limitation that only minimally detracts from the impressive overall scope of the study.

We agree. We have added to the discussion: “A full landscape of mutational effects can help to predict drug response and guide small molecule design to counteract acquired resistance. The ability to define molecular mechanisms towards that goal will likely require more purposefully chosen chemical inhibitors and combinatorial mutational libraries to be maximally informative.”

**Reviewer #3 (Public review):**
Summary:In the manuscript 'Mapping kinase domain resistance mechanisms for the MET receptor tyrosine kinase via deep mutational scanning' by Estevam et al, deep mutational scanning is used to assess the impact of ~5,764 mutants in the MET kinase domain on the binding of 11 inhibitors. Analyses were divided by individual inhibitor and kinase inhibitor subtypes (I, II, I 1/2, and III). While a number of mutants were consistent with previous clinical reports, novel potential resistance mutants were also described. This study has implications for the development of combination therapies, namely which combination of inhibitors to avoid based on overlapping resistance mutant profiles. While one suggested pair of inhibitors with the least overlapping resistance mutation profiles was suggested, this manuscript presents a proof of concept toward a more systematic approach for improved selection of combination therapeutics. Furthermore, in a final part of this manuscript the data was used to train a machine learning model, the ESM-1b protein language model augmented with an XG Boost Regressor framework, and found that they could improve predictions of resistance mutations above the initial ESM-1b model.Strengths:Overall this paper is a tour-de-force of data collection and analysis to establish a more systematic approach for the design of combination therapies, especially in targeting MET and other kinases, a family of proteins significant to therapeutic intervention for a variety of diseases. The presentation of the work is mostly concise and clear with thousands of data points presented neatly and clearly. The discovery of novel resistance mutants for individual MET inhibitors, kinase inhibitor subtypes within the context of MET, and all resistance mutants across inhibitor subtypes for MET has clinical relevance. However, probably the most promising outcome of this paper is the proposal of the inhibitor combination of Crizotinib and Cabozantib as Type I and Type II inhibitors, respectively, with the least overlapping resistance mutation profiles and therefore potentially the most successful combination therapy for MET. While this specific combination is not necessarily the point, it illustrates a compelling systematic approach for deciding how to proceed in developing combination therapy schedules for kinases. In an insightful final section of this paper, the authors approach using their data to train a machine learning model, perhaps understanding that performing these experiments for every kinase for every inhibitor could be prohibitive to applying this method in practice.

We thank Reviewer 3 for their assessment of our manuscript (we are very happy to have it described as a tour-de-force!)

Weaknesses:This paper presents a clear set of experiments with a compelling justification. The content of the paper is overall of high quality. Below are mostly regarding clarifications in presentation.Two places could use more computational experiments and analysis, however. Both are presented as suggestions, but at least a discussion of these topics would improve the overall relevance of this work. In the first case it seems that while the analyses conducted on this dataset were chosen with care to be the most relevant to human health, further analyses of these results and their implications of our understanding of allosteric interactions and their effects on inhibitor binding would be a relevant addition. For example, for any given residue type found to be a resistance mutant are there consistent amino acid mutations to which a large or small or effect is found. For example is a mutation from alanine to phenylalanine always deleterious, though one can assume the exact location of a residue matters significantly. Some of this analysis is done in dividing resistance mutants by those that are near the inhibitor binding site and those that aren't, but more of these types of analyses could help the reader understand the large amount of data presented here. A mention at least of the existing literature in this area and the lack or presence of trends would be worthwhile. For example, is there any correlation with a simpler metric like the Grantham score to predict effects of mutations (in a way the ESM-1b model is a better version of this, so this is somewhat implicitly discussed).

Indeed we experimented with including these types of features in the XGBoost scheme (particularly residue volume change and distance) to augment the predictive power of the ESM model - see Figure 8 - figure supplement 1; however, we didn’t find them as significant. Therefore, the signal is likely very small and/or incorporated into the baseline ESM model.

Indeed, this discussion relates to the second point this manuscript could improve upon: the machine learning section. The main actionable item here is that this results section seems the least polished and could do a better job describing what was done. In the figure it looks like results for certain inhibitors were held out as test data - was this all mutants for a single inhibitor, or some other scheme? Overall I think the implications of this section could be fleshed out, potentially with more experiments.

Figure 8A and the methods section contain a very detailed explanation of test data. We have thought about it and do not have any easy path to improve the description, which we reproduce here:

“Experimental fitness scores of MET variants in the presence of DMSO and AMG458 were ignored in model training and testing since having just one set of data for a type I ½ inhibitor and DMSO leads to learning by simply memorizing the inhibitor type, without generalizability. The remaining dataset was split into training and test sets to further avoid overfitting (Figure 8A). The following data points were held out for testing - (a) all mutations in the presence of one type I (crizotinib) and one type II (glesatinib analog) inhibitor, (b) 20% of randomly chosen positions (columns) and (c) all mutations in two randomly selected amino acids (rows) (e.g. all mutations to Phe, Ser). After splitting the dataset into train and test sets, the train set was used for XGBoost hyperparameter tuning and cross-validation. For tuning the hyperparameters of each of the XGBoost models, we held out 20% of randomly sampled data points in the training set and used the remaining 80% data for Bayesian hyperparameter optimization of the models with Optuna (Akiba et al., 2019), with an objective to minimize the mean squared error between the fitness predictions on 20% held out split and the corresponding experimental fitness scores. The following hyperparameters were sampled and tuned: type of booster (booster - gbtree or dart), maximum tree depth (max_depth), number of trees (n_estimators), learning rate (eta), minimum leaf split loss (gamma), subsample ratio of columns when constructing each tree (colsample_bytree), L1 and L2 regularization terms (alpha and beta) and tree growth policy (grow_policy - depthwise or lossguide). After identifying the best combination of hyperparameters for each of the models, we performed 10-fold cross validation (with re-sampling) of the models on the full training set. The training set consists of data points corresponding to 230 positions and 18 amino acids. We split these into 10 parts such that each part corresponds to data from 23 positions and 2 amino acids. Then, at each of 10 iterations of cross-validation, models were trained on 9 of 10 parts (207 positions and 16 amino acids) and evaluated on the 1 held out part (23 positions and 2 amino acids). Through this protocol we ensure that we evaluate performance of the models with different subsets of positions and amino acids. The average Pearson correlation and mean squared error of the models from these 10 iterations were calculated and the best performing model out of 8192 models was chosen as the one with the highest cross-validation correlation. The final XGBoost models were obtained by training on the full training set and also used to obtain the fitness score predictions for the validation and test sets. These predictions were used to calculate the inhibitor-wise correlations shown in Figure 8B.“

As mentioned in the 'Strengths' section, one of the appealing aspects of this paper is indeed its potential wide applicability across kinases -- could you use this ML model to predict resistance mutants for an entirely different kinase? This doesn't seem far-fetched, and would be an extremely compelling addition to this paper to prove the value of this approach.

This is exactly where we want to go next! But as we see here, it is going to be hard and require more purposeful selection of chemicals and likely combinatorial mutations to be maximally informative (see also reviewer 2 response where we have added text)

Another area in which this paper could improve its clarity is in the description of caveats of the assay. The exact math used to define resistance mutants and its dependence on the DMSO control is interesting, it is worth discussing where the failure modes of this procedure might be. Could it be that the resistance mutants identified in this assay would differ significantly from those found in patients? That results here are consistent with those seen in the clinic is promising, but discrepancies could remain.

Thank you for pointing this out. The greatest trade-off of probing the intracellular MET kinase (juxtamembrane, kinase domain, c-tail) in the constitutively active TPR system is that while we gain cytoplasmic expression, constitutive oligomerization, and HGF-independent activation, other features like membrane-proximal effects are lost and translatability of some mutations in non-proliferative conditions may also be limited. Nevertheless, Ba/F3 allows IL-3 withdrawal to serve as an effective variant readout of transgenic kinase variant effects due to its undetectable expression of endogenous RTKs and addiction to exogenous interleukin-3 (IL-3).

In our previous study, we were also interested in comparing the phenotypic results to available patient populations in cBioPortal. We observed that our DMS captured known oncogenic MET kinase variants, in addition to a population of gain-of-function variants within clinical residue positions that have not been clinically reported. Interestingly, the population of possible novel gain-of-function mutant codons were more distant in genetic space (2-3 Hamming distance) from wild type than the clinically reported variant codon (1-2 Hamming distance).

For this inhibitor screen, we also carefully compared previously reported and validated resistance mutations across referenced publications to that of our inhibitor screen, and observed large agreement as noted in-text. While discrepancies could definitely remain, there is precedence for consistency.

Furthermore a more in depth discussion of the MetdelEx14 results is warranted. For example, why is the DMSO signature in Figure 1 - supplement 4 so different from that of Figure 1?

In our previous study (Estevam et al., 2024), we more directly compared MET and METΔExon14, and while observed several differences, especially at conserved regulatory motifs, the TPR expression system did not provide a robust differential. Therefore, we hypothesize that a membrane-bound context is likely necessary to obtain a differential that captures juxtamembrane regulatory effects for these two isoforms. For that reason, we did not place heavy emphasis on the differences between MET and METΔExon14 in this study. Nevertheless, we performed parallel analysis of the METΔExon14 inhibitor DMS and provided all source and analyzed data in our GitHub repository (https://github.com/fraser-lab/MET_kinase_Inhibitor_DMS).

In our analysis of resistance, we used Rosace to score and compare DMSO and inhibitor landscapes. We present the full distribution of raw scores in Figure 1 for each condition. However, to visually highlight resistance mutations as a heatmap, we subtracted the scores of each variant in each inhibitor condition from the raw DMSO score, making the heatmaps in Figure 1 - supplement 4 appear more “blue.”

And finally, there is a lot of emphasis put on the unexpected results of this assay for the tivantinib "type III" inhibitor - could this in fact be because the molecule "is highly selective for the inactive or unphosphorylated form of c-Met" according to Eathiraj et al JBC 2011?

The work presented by Eathiraj et al JBC 2011 is a key study we reference and is foundational to tivantinib. While the point brought up about tivantinib’s selective preference for an inactive conformation is valid, this is also true for type II kinase inhibitors. In our study, regardless of inhibitor conformational preference, tivantinib was the only one with a nearly identical landscape to DMSO and exhibited selection even in the absence of Ba/F3 MET-addiction (Figure 1E). This result is in closer agreement with MET agnostic behavior reported by Basilico et al., 2013 and Katayama et al., 2013.

While this paper is crisply written with beautiful figures, the complexity of the data warrants a bit more clarity in how the results are visualized. Namely, clearly highlighting mutants that have previously reported and those identified by this study across all figures could help significantly in understanding the more novel findings of the work.

To better compare and contrast novel mutation identified in this study to others, we compiled a list of reported resistance mutations from recent clinical and experimental studies (Pecci et al 2024; Yao et al., 2023; Bahcall et al., 2022; Recondo et al., 2020; Rotow et al ., 2020; Fujino et al., 2019), since a direct database with resistance annotations does not exist for MET, to the best of our knowledge. In total, this amounted to 31 annotated resistance mutations across crizotinib, capmatinib, tepotinib, savolitinib, cabozantinib, merestinib, and glesatinib, which we have now tabulated in a new figure (Figure 4) and commentary in the main text:

To assess the agreement between our DMS and previously annotated resistance mutations, we compiled a list of reported resistance mutations from recent clinical and experimental studies (Pecci et al 2024; Yao et al., 2023; Bahcall et al., 2022; Recondo et al., 2020; Rotow et al ., 2020; Fujino et al., 2019; Figure 4A,B). Overall, previously discovered mutations are strongly shifted to a GOF distribution for the drugs where resistance is reported from treatment or experiment; in contrast, the distribution is centered around neutral for those sites for other drugs not reported in the literature (Figure 4C). However, even in cases such as L1195V, we observe GOF DMS scores indicative of resistance to previously reported inhibitors. Given this overall strong concordance with prior literature and clinical results, we can also provide hypotheses to clarify the role of mutations that are observed in combination with others. For example, H1094Y is a reported driver mutation that has been linked to resistance in METΔEx14 for glesatinib with either the secondary L1195V mutation or in isolation (Recodo et al., 2020). However, in our assay H1094Y demonstrated slight sensitivity to gelesatinib, suggesting that either resistance is linked to the exon14 deletion isoform, the L1195V mutation, or a cellular factor not modeled well by the BaF3 system.

Finally, the potential impacts and follow-ups of this excellent study could be communicated better - it is recommended that they advertise better this paper as a resource for the community both as a dataset and as a proof of concept. In this realm I would encourage the authors to emphasize the multiple potential uses of this dataset by others to provide answers and insights on a variety of problems.

Please see below

Related to this, the decision to include the MetdelEx14 results, but not discuss them at all is interesting, do the authors expect future analyses to lead to useful insights? Is it surprising that trends are broadly the same to the data discussed?

Our previous paper suggests that Ba/F3 isn’t a great model for measuring the differences between MET and METΔEx14, so we haven’t emphasized other than to point to our previous paper. We include the full analysis here nonetheless as a resource. Potentially where the greatest differences between resistance mutant behaviors would be observed is in the full-length, membrane-bound MET and METΔEx14 receptor isoforms. While outside of the scope of this study, there is great potential to use the resistance mutations identified in this study as a filtered group to test and map differential inhibitor sensitivities between receptor isoforms.

And finally it could be valuable to have a small addition of introspection from the authors on how this approach could be altered and/or improved in the future to facilitate the general application of this approach for combination therapies for other targets.

See also reviewer 2 response where we have added text.

**Recommendations for the authors:**

**Reviewer #1 (Recommendations for the authors):**
Major points of revision:(1) It seems like much of the structural interpretation of the inhibitor binding mode, outside of crizotinib binding, appears to come from docked models of the inhibitor to the MET kinase domain. Given the potential variability of the docked structure to the kinase domain, it would be useful for the authors to consider alternative possible binding modes that their docking pipeline may have suggested. It could also be useful to provide some degree of validation or contextualization of their docking models.

All individual figures are very carefully inspected based on either existing crystal structures of the inhibitor or closely related inhibitors (ATP, 3DKC; crizotinib, 2WGJ; tepotinib, 4R1V; tivantinib, 3RHK; AMG-458, 5T3Q; NVP-BVU972, 3QTI; merestinib, 4EEV; savolitinib, 6SDE). In total, four structural interpretations were the result of docking onto reference experimental structures (capmatinib, cabozantinib, glumetinib, glesatinib). As we wrote above, different conformations and binding modes are possible in predicted mutant structures (as we did here at scale) and included in the ML analysis already.

(2) In the first section, the authors classify an inhibitor as Type Ia on docking models, but mention the conflicting literature describing it as type Ib - it would be helpful to provide a contextualization of why this distinction between Ia and Ib matters, and what difference it might make. It would also be useful to know if their docking score only suggested poses compatible with Ia or if other poses were provided as well. Validation using other method might be beneficial, especially since they acknowledge the conflicting literature for classification. Or at least recontextualization that more evidence would be needed.

Kinase inhibitors have several canonical structural definitions we use to base the classifications in this study. Specifically, type I inhibitors are classified in MET by interactions with Y1230, D1228, K1110 in addition to its conformation in the ATP-binding site. Type I inhibitors are further subdivided into type 1a in MET if it leverages interactions with the solvent front and residue G1163. In prior literature referenced, tepotinib was classified as type 1b, which would imply it does not have solvent front interactions, like savolitinib (PDB 6SDE) or NVP-BVU972 (PDB 3QTI). However, in the tepotinib experimental structure (PDB 4R1V), we observed a greater structural resemblance to other type 1a inhibitors opposed to type 1b (Figure 1 - figure supplement 1b).

(3) The measure used to discuss resistance and sensitivity is ultimately a resistance score derived from the increase or decrease of the presence of a variant during cell growth. This is not a measure of direct binding. It would be helpful if the authors discussed alternative mechanisms through which these variants may impact resistance and/or sensitivity, such as stability, protonation effects, or kinase activity. The score itself may be convolving over all these potential mechanisms to drive GOF and LOF observed behavior.

See the response to the public review. Indeed, our ML framework explicitly included conformational and stability effects as significant in improving predictions.

(4) While it is promising to try and improve the predictive properties of ESM1b, it is not exactly clear why the authors considered their structural data of 11 inhibitors a sufficient dataset with which to augment the model. It would be useful for the authors to provide some additional context for why they wished to augment ESM1b in particular with their dataset, and provide any metrics indicating that their training data of 11 inhibitors provided an adequate statistical sample.

We don’t understand what this means. Sorry!

(5) The authors use ESM-1b to predict the fitness impact of each mutation and augment it using protein structural data of drug-target interactions. However, using an XGBoost regressor on a single set of 11 kinase-inhibitor interaction pairs is an incredibly sparse dataset to train upon. It would be useful for the authors to consider the limitations of their model, as well as its extensibility in the context of alternate binding poses, alternate conformations, or changes in protonation states of ligand or inhibitor.

On the contrary - this is 11 chemicals across 3000 mutations. We have discussed alternative interpretations above.

Minor points:(1) It would also be useful for the authors to provide more context around their choice of regressor. XGBoost is a powerful regressor but can easily overfit high dimensional data when paired with language models such as ESM-1b. This would be particularly useful since some of the features to train on were also generated using existing models such as ThermoMPNN.

Yes - we are quite concerned about overfitting and have tried to assess overfitting by careful design of test and validation sets.

(2) The authors also mention excluding their DMSO and AMG458 scores in the model training and testing due to overfitting issues - it would be useful to have an SI figure pointing to this data.

No - we exclude the DMSO because that is the reference (baseline) and AMG because it has a different binding mode. This isn’t related to overfitting.

(3) The authors mention in their docking pipeline that 5 binding modes were used for each ligand docking, but it appears that only one binding mode is considered in the main figures. It would be useful for the authors to provide additional details about what were the other binding modes used for, how different were each binding mode, and how was the "primary" mode selected (and how much better was its score than the others).

The reviewer misinterprets the difference between poses shown in figures, based on mostly crystal structures or carefully selected templates, and the use of docked models in feature engineering for the ML part of the study. Where existing crystal structures do not exist, we performed docking for capmatinib, cabozantinib, glumetinib, glesatinib onto reference structures bound to type I (2WGJ) and type II (4EEV) inhibitors. We selected one representative binding mode based on the reference inhibitor, and while not exact, at a minimum these models provide a basis for structural interpretation.

**Reviewer #2 (Recommendations for the authors):**
My main suggestion is for the authors to add a few sentences (in non-technical language) to the results section, specifically before the results shown in Figure 3, defining gain-of-function, loss-of-function, resistance, and sensitivity. While these definitions are present in the materials and methods section, explicitly discussing them prior to the relevant results would significantly improve the overall readability of the manuscript.

We defined “gain-of-function” and “loss-of-function” mutations as those with fitness scores statistically greater or lower than wild-type. Within the DMSO condition, gain-of-function and loss-of -function labels describe mutational perturbation to protein function, whereas within inhibitor conditions, the labels describe the difference in fitness introduced by an inhibitor.

We have also clarified these definitions where the terms are first introduced: “As expected, the DMSO control population displayed a bimodal distribution with mutations exhibiting wild-type fitness centered around 0, with a wider distribution of mutations that exhibited loss- or gain-of-function effects, as defined by fitness scores with statistically significant lower or greater scores than wild-type, respectively.”

Figure 7D. Please add a bit more detail to the legend on how fold change (y-axis) was calculated.

Here, fold change represents the number of viable cells at each inhibitor concentration relative to the TKI control, measured with the CellTiter-Glo Luminescent Cell Viability Assay (Promega) as an end point readout. We have updated the legend of Figure 7D with calculation details: “Dose-response for each inhibitor concentration is represented as the fraction of viable cells relative to the TKI free control.”

I must admit, I did not understand what "Specific inhibitor fitness landscapes also aid in identifying mutations with potential drug sensitivity, such as R1086 and C1091 in the MET P-loop" means. These are positions where most mutations lead to greater sensitivity to crizotinib. Is the idea that there are potentially clinically-relevant MET mutations that can be targeted over wild type with crizotinib?

Thank you for highlighting this! The P-loop (phosphate-binding loop) is a glycine-rich structural motif conserved in kinase domains. This motif is located in the N-lobe, where its primary role is to gate ATP entry into the active site and stabilize the phosphate groups of ATP when bound. Therefore, the P-loop is a common target region for ATP-competitive inhibitor design, but also a site where resistance can emerge (Roumiantsev et al., 2002). The idea we’d like to convey is that identifying residues that offer the potential for drug stabilization with the added benefit of having lower risk resistance, is an attractive consideration for novel inhibitor design.

We have added to the text: “Individual inhibitor resistance landscapes also aid in identifying target residues for novel drug design by providing insights into mutability and known resistance cases. This enables the selection of vectors for chemical elaboration with potential lower risk of resistance development. Sites with mutational profiles such as R1086 and C1091, located in the common drug target P-loop of MET, could be likely candidates for crizotinib.”

**Reviewer #3 (Recommendations for the authors):**
(1) Suggested Improvements to the Figures:a) Figure 4A - T1261 seems to be mislabeledb) In Figure 3A it's suggested to highlight mutants determined to be resistance mutants by this scheme.c) In Figure 3D it would be informative to highlight which of these resistance mutants have already been previously reported and which are novel to this studyd) Throughout figures 3A, 3D, and 4G the graphical choices on how to highlight synonymous mutations and mutations not performed in the assay needs improvement.The Green vs Grey 'TRUE' vs 'FALSE' boxes are confusing. Just a green box indicating synonymous mutations would be sufficient. Additionally these green boxes are hard to see, and often edges of this green box are currently missing making it even more difficult to see and interpret.* In Figure 4A mutants do not seem to be indicated by a line or plus sign, but this is not explained in the legend or the caption. Please add.* In 3D and 4G it is not clear if the mutants not performed are indicated at all - perhaps they are indicated in white, making them indistinguishable from scores with 0. Please clarify.

T1261 and G1242 are now correctly labeled.

In text we have also highlighted reported resistance mutations for crizotinib, which are inclusive of clinical reports and in vitro characterization: “These sites, and many of the individual mutations, have been noted in prior reports, such as: D1228N/H/V/Y, Y1230C/H/N/S, G1163R.”

We have adjusted the heatmaps to improve visual clarity. Mutations with score 0 are white, as indicated by the scale bar, and mutations uncaptured by the screen are now in light yellow. The green outline distinguishing WT synonymous mutations have also been adjusted so edges are no longer cut off. In our representations, we only distinguished mutations by the score color scale bar and WT outline. What looked like a “plus” or “line” in the original figure was only the heatmap background, which now should be resolved in the updated figure and legends for Figure 3 and Figure 4.

(2) Some Minor Suggested Improvements to the Text:a) The abbreviation CBL for 'CBL docking site' is used without being defined.b) Figure 3G is referenced, but it does not exist.c) In the sentence 'Beyond these well characterized sites, regions with sensitivity occurred throughout the kinase, primarily in loop-regions which have the greatest mutational tolerance in DMSO, but do not provide a growth advantage in the presence of an inhibitor (Figure 1 - Figure Supplement 1; Figure 1 - Figure Supplement 2).'. It is not clear why these supplemental figures are being referenced.d) In the supplement section 'Enrich2 Scoring' has what seem like placeholders for citations in [brackets]

Cbl is a E3 ubiquitin ligase that plays a role in MET regulation through engagement with exon 14, specifically at Y1003 when phosphorylated. This mode of regulation was more highlighted in our previous study. However, since Cbl was only mentioned briefly in this study, we have removed reference to it to simplify the text.

In addition, we have removed the figure 3G reference and corrected the in-text range. We have also removed references to figure supplements where unnecessary and edited the “Enrich2 scoring” method section to now reference missing citations.